# Genomic Constellation of Human Rotavirus G8 Strains in Brazil over a 13-Year Period: Detection of the Novel Bovine-like G8P[8] Strains with the DS-1-like Backbone

**DOI:** 10.3390/v15030664

**Published:** 2023-03-01

**Authors:** Roberta Salzone Medeiros, Yasmin França, Ellen Viana, Lais Sampaio de Azevedo, Raquel Guiducci, Daniel Ferreira de Lima Neto, Antonio Charlys da Costa, Adriana Luchs

**Affiliations:** 1Enteric Diseases Laboratory, Virology Center, Adolfo Lutz Institute, Sao Paulo 01246-902, Brazil; 2General Coordination of Public Health Laboratories, Department of Strategic Articulation in Epidemiology and Health Surveillance, Ministry of Health, Brasília 70068-900, Brazil; 3Medical Parasitology Laboratory (LIM/46), São Paulo Tropical Medicine Institute, University of Sao Paulo, Sao Paulo 05403-000, Brazil

**Keywords:** genomic analysis, rotavirus A, emergence, gastroenteritis, G8, bovine–human reassortants

## Abstract

Rotavirus (RVA) G8 is frequently detected in animals, but only occasionally in humans. G8 strains, however, are frequently documented in nations in Africa. Recently, an increase in G8 detection was observed outside Africa. The aims of the study were to monitor G8 infections in the Brazilian human population between 2007 and 2020, undertake the full-genotype characterization of the four G8P[4], six G8P[6] and two G8P[8] RVA strains and conduct phylogenetic analysis in order to understand their genetic diversity and evolution. A total of 12,978 specimens were screened for RVA using ELISA, PAGE, RT-PCR and Sanger sequencing. G8 genotype represented 0.6% (15/2434) of the entirely RVA-positive samples. G8P[4] comprised 33.3% (5/15), G8P[6] 46.7% (7/15) and G8P[8] 20% (3/15). All G8 strains showed a short RNA pattern. All twelve selected G8 strains displayed a DS-1-like genetic backbone. The whole-genotype analysis on a DS-1-like backbone identified four different genotype-linage constellations. According to VP7 analysis, the Brazilian G8P[8] strains with the DS-1-like backbone strains were derived from cattle and clustered with newly DS-1-like G1/G3/G9/G8P[8] strains and G2P[4] strains. Brazilian IAL-R193/2017/G8P[8] belonged to a VP1/R2.XI lineage and were grouped with bovine-like G8P[8] strains with the DS-1-like backbone strains detected in Asia. Otherwise, the Brazilian IAL-R558/2017/G8P[8] possess a “Distinct” VP1/R2 lineage never previously described and grouped apart from any of the DS-1-like reference strains. Collectively, our findings suggest that the Brazilian bovine-like G8P[8] strains with the DS-1-like backbone strains are continuously evolving and likely reassorting with local RVA strains rather than directly relating to imports from Asia. The Brazilian G8P[6]-DS-1-like strains have been reassorted with nearby co-circulating American strains of the same DS-1 genotype constellation. However, phylogenetic analyses revealed that these strains have some genetic origin from Africa. Finally, rather than being African-born, Brazilian G8P[4]-DS-1-like strains were likely imported from Europe. None of the Brazilian G8 strains examined here exhibited signs of recent zoonotic reassortment. G8 strains continued to be found in Brazil according to their intermittent and localized pattern, thus, does not suggest that a potential emergence is taking place in the country. Our research demonstrates the diversity of G8 RVA strains in Brazil and adds to the understanding of G8P[4]/P[6]/P[8] RVA genetic diversity and evolution on a global scale.

## 1. Introduction

Rotavirus A (RVA) is the leading cause associated with viral acute gastroenteritis in children worldwide, accounting for nearly 128,500 under-five deaths annually, even with the increasing implementation of universal RVA vaccination [1]. To date, over 100 countries have incorporated RVA vaccines into their national immunization programs (NPIs) [2]. Brazil introduced the Rotarix^TM^ vaccine into the NPI in 2006. Since then, a significant reduction in diarrhea-associated hospitalizations and death associated with infantile gastroenteritis has been observed [3]. 

RVA belongs to *Rotavirus* genus, *Sedoreovirinae* subfamily, *Reoviridae* family and *Riboviria* realm. The RVA genome contains eleven double-stranded RNA (dsRNA) segments that encode six structural proteins (VP1-VP4, VP6 and VP7) and six nonstructural proteins (NSP1-NSP5/6), surrounded by a three-layer capsid. The outer capsid proteins, VP7 (capsid glycoprotein) and VP4 (spike protein) independently elicit neutralizing antibodies and form the basis of the binary classification system of G and P types, respectively [4]. Globally, six RVA genotypes, G1P[8], G2P[4], G3P[8], G4P[8], G9P[8] and G12P[8], are commonly associated with human infections, accounting for about 90% of infections requiring medical attention [5]. Distinct G and P genotypes, such as G5, G6, G8, G10, G11, P[1], P[5], P[7], P[9] and P[14], have been sporadically detected in humans and are supposed to have originated from animal hosts through interspecies transmission events [6,7,8].

A whole-genome classification system is also used to assign genotypes to each segment, where Gx-P[x]-Ix-Rx-Cx-Mx-Ax-Nx-Tx-Ex-Hx represents the genotypes of VP7-VP4-VP6-VP1-VP2-VP3-NSP1-NSP2-NSP3-NSP4-NSP5/6, respectively [4]. The majority of human RVA strains possess Wa-like (genogroup 1) and DS-1-like (genogroup 2) genotype constellation. The AU-1-like (genogroup 3) genotype constellation is less common and rarely found in RVA human strains. G1P[8], G3P[8], G4P[8], G9P[8] and G12P[8] combinations are expressed on the Wa-like backbone; G2P[4] and G3P[9] on the DS-1-like and AU-1-like backbone, respectively [4]. Unusual human G/P combinations (i.e., G5P[6], G9P[23]) tend to possess more diverse genetic backbones [9,10]. Recently, an emergence of novel human intergenogroup ressortant strains, DS-1-like G1/G3/G9/G8P[8], were reported globally, including in Brazil [11,12,13,14,15]. Moreover, some of these atypical emergent strains resulted from human/animal reassortment events, such as the equine-like G3P[8] DS-1-like and the bovine-like G8P[8] strains with the DS-1-like backbone [16,17].

G8 is a common G type found in cattle [18], but sporadically detected in humans [5,19,20,21]. Interestingly, the G8 genotype is particularly prevalent in the African continent and often detected in combinations with P[4], P[6] and P[8] [22,23]. G8 strains paired with either P[4], P[6] or P[8] genotypes were previously reported in Brazil [21,24,25,26,27,28]; however, their origins remain obscure. A phylogenetic analysis conducted with three gene segments by Luchs and Timenetsky showed that Brazilian G8P[6] strains display close genetic relationships to bovine G8, bat P[6] and I2 human RVA strains, suggesting potential interspecies transmission events occurring between multiple hosts [26].

It is well known that interspecies transmission and reassortment between human and animal RVA contribute significantly to the virus’ evolution. Additionally, the introduction of RVA vaccines into human populations may impose additional selective pressure on circulating RVA strains, possibly influencing their evolutionary rate and the capability of producing new RVA strains to diffuse worldwide [29]. Continued surveillance is needed to verify the effectiveness of the Rotarix^TM^ vaccine in Brazil, together with potential emergence of unusual genotypes [30]. The whole-genotype analysis of RVA G8P[4]/P[6]/P[8] strains detected worldwide, including in Brazil, may help unravel the true origin of these strains, as well as understand their ability to eventually evade vaccine immunity.

The aims of the present study were to monitor G8 infections in the Brazilian human population between 2007 and 2020, undertake the full-genotype characterization of four G8P[4], six G8P[6] and two G8P[8] RVA strains and conduct phylogenetic analysis in order to understand their genetic diversity and evolution. 

## 2. Materials and Methods

### 2.1. Sampling

This study was carried out with convenient surveillance specimens. Brazil is a continental-size country, and the Brazilian RVA Surveillance Program is funded by three collaborating institutes: (i) Evandro Chagas Institute, the national and regional reference center for RVA surveillance in the Northern region and part of the Northeastern region; (ii) Oswaldo Cruz Institute, the regional reference center for RVA surveillance in part of the Northeastern, Southeastern and Southern regions; and (iii) Adolfo Lutz Institute, the regional reference center for RVA surveillance in Midwest and part of the Southeastern and Southern regions. From 2007 to 2020, a total of 12,978 stool samples collected from patients with acute gastroenteritis were sent to the Enteric Diseases Laboratory of the Adolfo Lutz Institute, together with relevant age, gender and location data. 

### 2.2. Rotavirus Detection and Electropherotyping

RVA was detected using a commercial immunoenzymatic assay (RIDASCREEN^®^ Rotavirus, R-Biopharm AG, Darmstadt, Germany) and performed according to the manufacturer’s instructions. The RVA migration profiles were analyzed by PAGE followed by silver staining of gels [31].

### 2.3. Viral RNA Extraction and G/P Genotyping

Viral RNA was extracted from 10% fecal samples using a *Bio Gene DNA/RNA Viral* (Quibasa–Quimica Basica Ltd., Belo Horizonte, BH, Brazil), according to the manufacturer’s instructions, and subjected to G and P typing by multiplex reverse transcription-polymerase chain reaction (RT-PCR) with type-specific primers, following previously described protocols [32,33]. First-round amplicons of all G8 VP7 RVA-positive samples and their respective VP4 segments were selected for sequencing. PCR amplicons were sequenced using the BigDye Terminator v3.1 Cycle Sequencing Kit (Applied Biosystems, Foster City, CA, USA) with primers Beg9/End9 (1062 bp) and Con2/Con3 (876 bp). Dye-labeled products were sequenced in an ABI 3500 sequencer (Applied Biosystems, Foster City, CA, USA). Sequencing chromatograms were edited manually using Sequencher™ 4.7 software (Gene Codes Corporation, Ann Arbor, MI, USA). The genotype assignment was accomplished using Rotavirus A Genotype Determination–ViPR (https://www.viprbrc.org/brc/rva) to confirm the detected G8 genotype and identify VP4 specificity.

### 2.4. Nucleotide Sequencing of the G8 RVA Segments

Based on sample availability and viral load of VP7/VP4 amplicons, 12 G8 RVA strains were selected for investigation of the 11 gene segments (Table 1). RVA dsRNA was extracted from 10% fecal samples using *Bio Gene DNA/RNA Viral* (Quibasa–Quimica Basica Ltd., Belo Horizonte, BH, Brazil) according to the manufacturer’s instructions. RT-PCRs for the 11 gene segments were performed *in house* using primers described by Varghese et al. [34] (VP1, VP2 and VP3), Ramani et al. [35] (VP3), Wang et al. [36] (NSP1, NSP2, NSP4, NSP5 and VP6), Magagula et al. [37] (NSP2, NSP3, NSP4, NSP5, VP6 and VP7), Mijatovic-Rustempasic et al. [38] (NSP5), He et al. [39] (NSP1) and Gentsch et al. [32] (VP4) following the amplification protocol formerly define by Gouvea et al. [33]. All PCR products were loaded onto 1.5% agarose gel containing GelRed™ (Biotium, Fremont, CA, USA) along with a 100 bp molecular-sized ladder and viewed in a gel-documentation system. PCR amplicons were sequenced using a BigDye™ Kit v3.1 (Applied Biosystems, Foster City, CA, USA) with the same primer set used in the PCR reaction. Dye-labeled products were sequenced using an ABI 3500 sequencer (Applied Biosystems, Foster City, CA, USA). Sequencing chromatograms were edited manually using Sequencher™ 4.7 software (Gene Codes Corporation, Ann Arbor, MI, USA). Genotype assignment was accomplished using Rotavirus A Genotype Determination–ViPR (https://www.viprbrc.org/brc/rva).

### 2.5. Phylogenetic Analyzes

In order to assess more insightful information about the phylogenetic relationships of the G8 genotypes detected in this study, the near-full length of VP7, VP6 and NSP1-5/6 RVA sequences and partial VP1-4 RVA sequences obtained were aligned with a set of prototype sequences available in the GenBank database using the CLUSTAL W algorithm in the BioEdit Sequence Alignment Editor software, version 7.0.5.2 (Ibis Therapeutics, Carlsbad, CA, USA). A maximum-likelihood tree was constructed for each genome segment. The best substitution models were selected based on the corrected Akaike Information Criterion (AICc) value as implemented in MEGA X [40]. The models used in this study were General time reversible (GTR) +G +I (NSP1), Tamura 3-parameter (T92) +G +I (NSP2, NSP3, NSP4, VP1, VP3, VP6 and VP4 P[4]), T92 +G (NSP5/6 and VP7), T92 +I (VP4 P[8]), Tamura-Nei (TN93) +G (VP2) and Hasegawa-Kishino-Yano (HKY) +G +I (VP4 P[6]). The statistical significance at the branch point was calculated with 1000 pseudo-replicate datasets. For the designation of lineages, strains from GenBank were selected using lineages previously published by Agbemabiese et al. [41] for NSP1 to -5, VP1 to -3 and VP6, Gupta et al. [42] and Doan et al. [43] for VP4, and Silva-Sales et al. [21]. 

Nucleotide sequences determined in this study have been deposited in GenBank under the accession numbers ON653042, ON653043, ON677532-ON677537, ON703253, ON722359-ON722361 (NSP1), ON745820-ON745831 (NSP2), ON807575-ON807586 (NSP3), ON885866-ON885877 (NSP4), ON938328-ON938339 (NSP5), OP179787-OP179798 (VP1), OP232077-OP232088 (VP2), OP407951-OP407962 (VP3), OP374084-OP374095 (VP4), OP263659-OP263670 (VP6) and OP311907-OP311918 (VP7).

### 2.6. Antigenic and Structural Analysis of VP7 Gene Segment

Antigenic characterization sequences were aligned using BioEdit Sequence Alignment Editor software, version 7.0.5.2 (Ibis Therapeutics, Carlsbad, CA, USA), and potential N-linked glycosylation sites were screened using the NetNGlyc1.0 Server (https://services.healthtech.dtu.dk/service.php?NetNGlyc-1.0). 

The sequences obtained were aligned and converted to proteins using reference sequences to identify the open reading frame and subsequent protein alignment. With the sequences created in this manner, we separated each case into its own FASTA file and proceeded with the modeling in MODELER 10.4 and SwissModel, evaluating them with DOPE scores [44,45]. The sequences were then evaluated according to the PDBSum GENERATE scores [46] and structurally aligned by the PyMod modeler module (SAlign) from the Pymol 2.5 (https://pymol.org/2/). The models were then treated on the MolPorbity website (http://molprobity.biochem.duke.edu/) to check clashes and bumps. The final configurations were then evaluated on the Immune Epitope Database (IEDB) website (https://www.iedb.org/) in the DISCOTOPE and ELlipro modules to investigate discontinuous epitopes and predict antibody binding, both using 3D structures for the losses.

### 2.7. Ethical Approval

Previous Ethics Committee approval was granted by the Adolfo Lutz Institute, São Paulo, Brazil (CAAE 40718114.5.0000.0059 and CAAE 51963821.3.0000.0059). 

## 3. Results

### 3.1. RVA Detection and Genotyping

The G8 genotype represented 0.6% (15/2434) of the entirely RVA-positive samples detected between 2007 and 2020. All G8 VP7 and VP4 amplicons were successfully sequenced and further differentiated as G8P[4] (33.3%, 5/15), G8P[6] (46.7%, 7/15) and G8P[8] (20%, 3/15) combinations. All G8 strains showed a short RNA pattern. The short electrophoretic profile is commonly associated with P[4] and P[6] specificities, but unusually associated with those of P[8], suggesting a potential identification of the emerging DS-1-like G8P[8] strain (Table 1). A distribution of RVA genotypes detected during the study period was reported in previous investigations [13,14,30,47].

Complete or nearly complete nucleotide sequences for NSP1-5/6, VP7 and VP6 genome segments and partial VP1–4 gene sequences of 12 selected G8 strains were determined. The percentage of the genomes obtained ranged from 42.9% to 49.8%. The length of the sequences determined for the 12 G8 strains and the nucleotide positions compared are shown in Appendix A. The four G8P[4], six G8P[6] and two G8P[8] RVA strains possess a DS-1-like genetic background (I2-R2-C2-M2-A2-N2-T2-E2-H2), thus confirming the identification of the novel DS-1-like G8P[8] intergenogroup reassortment (Table 1).

To investigate the genetic relatedness and potential origin of the Brazilian G8P[4]/P[6]/P[8] strains, the 11 gene segments were analyzed phylogenetically. The phylogenetic relationship was inferred by the maximum-likelihood method, using reference RVA strains from humans, vaccines, cows, goats, foxes, alpacas, bats, pigs, camels, sheep, roe deer, horses, cats, vicuñas, simians, dogs, rabbits, antelopes, guanacos, rats, llamas and lambs available at the GenBank database. Sequences from Brazil and South America were also included in the analysis.

### 3.2. Bovine-like G8P[8] Strains with the DS-1-like Backbone

Appendix A shows a comparison of amino acid sequences of the six antigenic regions A–F [48] between G8P[8] and the DS-1-like backbone strains detected here and reference RVA strains belonging to the G8 genotype. There was 100% amino acid (aa) homology in all antigenic regions between RVA/Human-wt/BRA/R193/2017/G8P[8] and RVA/Human-wt/BRA/R558/2017/G8P[8] and the recently emerged human bovine-like G8P[8] strains and the DS-1-like backbone strains reported in Asia (RVA/Human-wt/THA/SKT-457/2014/G8P[8], RVA/Human-wt/VNM/RVN1293/2014/G8P[8] and RVA/Human-wt/JPN/MU14-0/2014/G8P[8]) in 2014 [17,49,50]. The Brazilian G8P[8] with the DS-1-like backbone strains detected here were also identical, compared to the six antigenic regions to the Argentinian G8P[8] strains bearing DS-1-like backbone (RVA/Human-wt/ARG/Arg15080/2016/G8P[8] and RVA/Human-wt/ARG/Arg16571/2018/G8P[8]) reported since 2016 [51]. They were also identical to the bovine RVA/Cow-wt/IND/68/2007/G8P[14] strain, thus supporting the animal origin hypothesis of the emerging bovine-like G8P[8] strains with the DS-1-like backbone. The alignment of aa sequences deduced from the VP7 gene revealed aa substitutions in Brazilian bovine-like G8P[8] DS-1-like strains inside the variable region A (aa 39–50) at positions 41^V→I^ and 44 ^V→I^, region B (aa 87–101) at position 87^A→T^ and region F (aa 233–242) at position 237^V→I^. Amino acid substitutions were also observed outside VP7 hypervariable regions at positions 65^M/T→A^, 119^K→R^ and 268^I→V^. The VP7 protein of Brazilian bovine-like G8P[8] DS-1-like strains had two potential N-linked glycosylation sites located at aa 69-72 (NVST) and 238-241 (NVTT).

Appendix A shows the deduced amino acid sequence of the VP4 (subunit VP8*) of human Brazilian bovine-like G8P[8] strains with the DS-1-like backbone (RVA/Human-wt/BRA/IAL-R193/2017/G8P[8] and RVA/Human-wt/BRA/IAL-R558/2017/G8P[8]) and representative VP4 amino acid sequences of the RVA P[8] genotype. The three potential cleavage sites, arginine (R) 230, 240 and 246, were maintained in the two Brazilian bovine-like G8P[8] strains with the DS-1-like backbone. The arginine at position 246 in Asian emergent bovine-like G8P[8] strains with the DS-1-like backbone RVA/Human-wt/JPN/MU14-0/2014/G8P[8], RVA/Human-wt/THA/SKT-457/2014/G8P[8], RVA/Human-wt/VNM/RVN1149/2014/G8P[8] were substituted by a lysine (K). The highly conserved cysteine (C), at residue 215, and prolines (P), at residues 68, 71, 224 and 225, were maintained in the two Brazilian bovine-like G8P[8] strains with the DS-1-like backbone. 

Within the VP8* subunit variable region, substitutions had occurred in strain RVA/Human-wt/BRA/IAL-R193/2017/G8P[8] at positions 162^R→K^ and 195^G/S→D^. Amino acid substitutions were also observed outside the VP4 hypervariable region in the RVA/Human-wt/BRA/IAL-R193/2017/G8P[8] strain: 35^V→I^, 236^G/S→D^ and 245^K→T^. There was 100% aa homology in the VP4 hypervariable region between the RVA/Human-wt/BRA/IAL-R193/2017/G8P[8] strain and the recently emerged human bovine-like G8P[8] strains, with the DS-1-like backbone reported in Asia (RVA/Human-wt/THA/SKT-457/2014/G8P[8], RVA/Human-wt/VNM/RVN1149/2014/G8P[8], RVA/Human-wt/THA/SKT-109/2013/G1P[8] and RVA/Human-wt/JPN/MU14-0/2014/G8P[8]) in 2014, as well as the Russian RVA G12P[8] detected in 2017 (RVA/Human-wt/RUS/NS17-A1039/2017/G12P[8]) [17,49,50]. Considering the RVA/Human-wt/BRA/IAL-R558/2017/G8P[8] strain, substitutions had occurred at positions 146^S→G^ and 162^R→K^ within the VP8* subunit variable region. Amino acid substitutions were also observed outside the VP4 hypervariable region in RVA/Human-wt/BRA/IAL-R558/2017/G8P[8] strain: 245^K→T^. RVA/Human-wt/BRA/R193/2017/G8P[8] strain was unique, considering aa homology in VP4 hypervariable region (Appendix A).

Analysis of the genomic constellation of Brazilian bovine-like G8P[8] strains with the DS-1-like backbone detected here revealed two different genotype lineage constellations: G8.IV-P[8].III-A2.IVa-N2.XV-T2.V-E2.XII-H2.IVa-R2.XI-C2.IVa-M2.V-I1.V, represented by the RVA/Human-wt/BRA/IAL-R193/2017/G8P[8] strain, and G8.IV-P[8].III-A2.IVa-N2.XV-T2.V-E2.XII-H2.IVa-R2.Distinct-C2.IVa-M2.V-I1.V, represented by the RVA/Human-wt/BRA/IAL-R558/2017/G8P[8] strain. On the one hand, the Brazilian IAL-R193 strain possesses a genotype lineage constellation identical to the bovine-like G8P[8] bearing DS-1-like backbone strains that emerged in and spread from Asia in 2014, including those from Japan (RVA/Human-wt/JPN/MU14-0/2014/G8P[8]), Thailand (RVA/Human-wt/THA/SKT-457/2014/G8P[8]) and Vietnam (RVA/Human-wt/VNM/RVN1149/2014/G8P[8]) [17,49,50]. On the other hand, the Brazilian IAL-R558 strain exhibits a potentially unique genotypic lineage constellation in which the VP1 gene was reassorted with a yet undescribed lineage, here named “Distinct”. The other 10 backbone segments of the IAL-R558 strain belong to the same lineages identified in the Brazilian IAL-R193 strain (Table 1).

The sequences of the two Brazilian bovine-like G8P[8] strains with the DS-1-like backbone strains were analyzed to elucidate the origin of these strains, whether they are derived from one specific bovine-like G8P[8] DS-1-like reference strain, or whether they are reassortment strains. The VP7, VP4 and NSP2 genes of strains IAL-R193 and IAL-558 clustered together exclusively with bovine-like G8P[8] strains with the DS-1-like backbone circulating in Asian countries (nucleotide sequence identities of 99–100%, 98–99% and 99.5–100%, respectively) (Figure 1A,B,J). Considering the VP7 gene in particular, the two bovine-like G8P[8] strains with the DS-1-like backbone clustered together with the majority of human RVA G8P[8] DS-1-like reference strains and with the bovine RVA/Cow-wt/IND/68/2007/G8P[14] ancestor, reinforcing the hypothesis that these newly discovered bovine-like G8P[8] strains with the DS-1-like backbone have an animal origin (Figure 1A). Considering the VP4 gene, the Brazilian bovine-like G8P[8] strains with the DS-1-like backbone detected here grouped into Lineage III, as expected. After 2003, virtually all globally circulating P[8] strains belonged to Lineage III (Figure 1B).

On the one hand, the VP2, VP3, VP6 and NSP4 phylogenetic analysis, strains IAL-R193 and IAL-R558 formed clusters with the Asian bovine-like G8P[8] strains with the DS-1-like backbone, but also with Asian equine-like G3P[8] DS-1-like strains and Asian emerging double-gene reassorted G1/G9P[8] DS-1-like RVA strains (nucleotide sequence identities of 98.4–99.8%, 99.3–100%, 99.7–100% and 98.8–99.5%, respectively) (Figure 1F–H,L). On the other hand, the NSP1 and NSP3 genes of strains IAL-R193 and IAL-558 formed clusters with Asian, European and American intergenogroup reassorted DS-1-like G1/G3/G9/G8P[8] strains (nucleotide sequence identities of 98.4–99.7% and 99.2–100%, respectively) (Figure 1I,K). Considering the NSP5 gene analysis in particular, the Brazilian RVA/Human-wt/BRA/IAL-R193/2017/G8P[8] strain exhibited a close genetic relationship to those of Asian, European and American intergenogroup reassorted DS-1-like G3/G9/G8P[8] strains (99–99.8% nt), while the RVA/Human-wt/BRA/IAL-R558/2017/G8P[8] strain displayed a genetic relationship with Asian and Australian G2P[4] strains besides the Asian, European and American intergenogroup reassorted DS-1-like G3/G9/G8P[8] strains (99.5–100% nt) (Figure 1M). As highlighted in the genotype lineage constellations data, a key observation was extracted from the phylogenetic analysis of the VP1 gene, in which the RVA/Human-wt/BRA/IAL-R193/2017/G8P[8] strain is closely related to the Asian bovine-like G8P[8] strains with the DS-1-like backbone and double-gene reassorted G9P[8] DS-1-like strains (99.3–99.8% nt), whereas RVA/Human-wt/BRA/IAL-R558/2017/G8P[8] strain could not be placed in any lineage previously proposed by Agbemabiese et al. [41] (Figure 1E).

### 3.3. G8P[4] DS-1-like Strains

There was 100% aa homology in six VP7 antigenic regions (A–F) between RVA/Human-wt/BRA/IAL-R2597/2010/G8P[4], RVA/Human-wt/BRA/IAL-R2598/2010/G8P[4], RVA/Human-wt/BRA/IAL-R2600/2010/G8P[4] and RVA/Human-wt/BRA/IAL-R2601/2010/G8P[4] strains and the RVA/Human-wt/GER/GER1H-09/2009/G8P[4] strain detected in Germany in 2009 [52]. There was also 100% in 5 VP7 gene antigenic regions (B–F) between Brazilian G8P[4] DS-1-like strains reported here and the human RVA/Human-wt/CIV/6736/2004/G8P[8], RVA/Human-wt/SVN/SI-885/2006/G8P[8] and RVA/Human-wt/ITA/SS65/2011/G8P[4] strains detected in the Ivory Coast in 2004, Slovenia in 2006 and Italy in 2011, respectively [53,54,55]. The alignment of aa sequences deduced from the VP7 gene revealed aa substitutions in G8P[4] strains inside the variable region A (aa 39–50) at positions 43^I→V^, 44^V→I^ and 45^T→A^, region B (aa 87–101) at positions 87^A/V→T^ and 100^D→E^, region C (aa 120–130) at position 122^T/V→A^, region D (aa 143–152) at position 145^N→S^, region E (aa 207–2020) at position 218^I→V^ and region F (aa 233–242) at position 237^V→I^. Amino acid substitutions were also observed outside VP7 hypervariable regions at positions 16^L→P^, 29^I→V^, 65^M→T^, 72^T/N→A^, 73^S/P→Q^, 100^D→E^ and 268^I→V^. The VP7 protein of Brazilian G8P[4] strains had two potential N-linked glycosylation sites located at aa 69–72 (NVSA) and 238–241 (NVTT) (Appendix A). 

Appendix A shows the deduced amino acid sequence of the VP4 (subunit VP8*) of human Brazilian G8P[4] DS-1-like strains (RVA/Human-wt/BRA/IAL-R2597/2010/G8P[4], RVA/Human-wt/BRA/IAL-R2598/2010/G8P[4], RVA/Human-wt/BRA/IAL-R2600/2010/G8P[4] and RVA/Human-wt/BRA/IAL-R2601/2017/G8P[4]) as well as representative VP4 amino acid sequences of the RVA P[4] genotype. The three potential cleavage sites, arginine (R)^®^ 230, 240 and 246, were maintained in the four Brazilian G8P[4] DS-1-like strains as well as in all reference strains. The highly conserved cysteine (C) at residue 215 and prolines (P) at residues 68, 71, 224 and 225 were also maintained in the four Brazilian G8P[4] DS-1-like strains. Within the VP4 hypervariable region, the Brazilian G8P[4] DS-1-like strains are virtually identical to the European RVA/Human-wt/GER/GER1H-09/2009/G8P[4] and RVA/Human-wt/ITA/SS65/2011/G8P[4] strains, except for one amino acid substitution that occurred in the Brazilian G8P[4] DS-1-like strains at position 166^V→M^. General amino acid substitutions were also observed at positions 87^N→S^, 149^S→G^, 162^R→G^, 166^V→M^ and 191^A→T^ inside the VP4 hypervariable region. 

Gene amplification using RT-PCR, nucleotide sequencing and sequence analysis of amplicons revealed that the four Brazilian G8P[4] DS-1-like strains described here had a G8.I-P[4].IVa-A2.II-N2.X-T2.V-E2.VII-H2.IVa-R2.V-C2.IVa-M2.V-I1.V genotype lineage constellations. The RVA/Human-wt/DEU/GER1H-09/2009/G8P[4] strain shares the same lineage genotype lineage constellation. It is likely that the RVA/Human-wt/ITA/SS65/2011/G8[4] strain also carries the same lineage genotype constellation; however, the lack of VP1, VP2 and VP3 nucleotide segments hampered the comparison (Table 1). 

The phylogenetic analysis of the 11 gene segments confirmed the close genetic relationship between the four Brazilian G8P[4] DS-1-like strains and the RVA/Human-wt/DEU/GER1H-09/2009/G8P[4] and RVA/Human-wt/ITA/SS65/2011/G8[4] strains. The VP1-4, VP6-7 and NSP1-5/6 genes from the four Brazilian G8P[4] DS-1-like strains (RVA/Human-wt/BRA/IAL-R2601/2010/G8P[4], RVA/Human-wt/BRA/IAL-R2600/2010/G8P[4], RVA/Human-wt/BRA/IAL-R2598/2010/G8P[4] and RVA/Human-wt/BRA/IAL-R2597/2010/G8P[4]) exhibited a high level of sequence conservation, with >99.4% sequence identity to each other (Figure 1A,B,E–M). Percentages of nt identity between the Brazilian G8P[4] DS-1-like, the RVA/Human-wt/DEU/GER1H-09/2009/G8P[4] and RVA/Human-wt/ITA/SS65/2011/G8[4] strains were remarkably similar across 10 RVA gene segments: 99.5–99.8% for NSP1, 99.6–100% for NSP2, 99.6–99.9% for NSP3, 99.5–99.8% for NSP4, 99.7–99.8% for NSP5, 99.5–99.7% for VP1, 99.5% for VP2, 99.5–99.3% for VP3, 99.7% for VP4 and 99.6–99.8% for VP6 (Figure 1C,E–M). 

With respect to the VP7 gene segment, an exception was observed. Brazilian G8P[4] DS-1-like IAL-R2600, IAL-R2601, IAL-R2597 and IAL-R2598 strains exhibited close genetic connection to those of G8P[4] strains in Germany (GER1H-09/2009) and Italy (SS65/2011), but also to G8P[4] strains previously reported in Brazil (TO-251/2010, IP-447MG/2011 and MA19555-11/2011) (98.6–100% nt) (Figure 1A). In fact, a key observation was extracted from the RVA/Human-wt/BRA/TO-251/2010/G8P[4] strain. The full-genome sequence of the RVA/Human-wt/BRA/TO-251/2010/G8P[4] strain was recently reported during genomic constellation surveillance conducted in the Brazilian state of Tocantins [21]. All five Brazilian G8P[4] strains (the four G8P[4] DS-1-like strains reported here and the TO-251/2010 strain) were detected in 2010 in the central part of the country, which is occupied by the Brazilian savanna biome. The analysis indicates that the four G8P[4] DS-1-like strains detected in this study clustered in distinct NSP1, NSP2 and NSP3 lineages separated from the RVA/Human-wt/BRA/TO-251/2010/G8P[4] strain represented by G8.I-P[4].IVa-A2.IVa-N2.V-T2.V-E2.V-H2.IVa-R2.V-C2.IVa-M2.V-I1.V genotype lineage constellations (Table 1). 

Brazilian G8P[4] DS-1-like (IAL-R2597, IAL-R2598, IAL-R2600 and IAL-R2601) strains also shared moderate high VP7 nucleotide identities with the bovine RVA G8P[5] Amasya-1/2015 (97.5–98.1% nt) strain isolated in Turkey, emphasizing once again the possible bovine origin of the G8 strains (Figure 1A). Additionally, a genetic analysis of the NSP2 gene revealed that the four human Brazilian G8P[4] DS-1-like strains also clustered together with the animal RVA/Sheep-tc/ESP/OVR762/2002/G8P[14] strain inside Lineage X, exhibiting sequences identities ranging from 96.7% to 96.9% (Figure 1J). A comparison of the Brazilian E2 NSP4 G8P[4] DS-1-like sequences showed that they were also related to the dog RVA strain ANK-K1 identified in Turkey in 2010, sharing the same Lineage VII and an identity of 94.4–94.5% at the nucleotide level (Figure 1L).

### 3.4. G8P[6] DS-1-like Strains

The five Brazilian G8P[6] DS-1-like strains characterized in the present study (RVA/Human-wt/BRA/IAL-RN361/2009/G8P[6], RVA/Human-wt/BRA/IAL-RN373/2009/G8P[6], RVA/Human-wt/BRA/IAL-RN374/2009/G8P[6], RVA/Human-wt/BRA/IAL-RN377/2009/G8P[6] and RVA/Human-wt/BRA/IAL-R2404/2010/G8P[6]) showed 100% aa homology in five VP7 gene antigenic regions (A–B and D–F) compared to the RVA/Human-wt/COD/KisB554/2010/G8P[6] strains and in four antigenic regions (A and D–F) compared to the RVA/Human-wt/COD/DRC86/2003/G8P[6] and RVA/Human-wt/COD/DRC88/2003/G8P[8] strains, all of which were African RVA G8 strains detected in Democratic Republic of Congo [56,57]. The alignment of aa sequences deduced from the VP7 gene revealed aa substitutions in the five Brazilian G8P[6] DS-1-like strains inside the variable region A (aa 39–50) at position 41^I→V^, region B (aa 87–101) at positions 87^T/V→A^ and 96^S→N^, region C (aa 120–130) at positions 122^T/A→V^ and 124^I→V^, region D (aa 143–152) at position 146^A→T^, region E (aa 207–2020) at position 218^V→I^ and region F (aa 233–242) at position 237^I→V^. Amino acid substitutions were also observed outside VP7 hypervariable regions at positions 21^I→V^, 65^T/A→M^, 72^T/A→N^, 116^V→I^, 139^I→V^, 186^S→A^ and 268^V→I^. The VP7 protein of the five Brazilian G8P[6] strains had two potential N-linked glycosylation sites located at aa 69–72 (NVSN) and 238–241 (NVTT) (Appendix A). 

The RVA/Human-wt/BRA/IAL-R2437/2010/G8P[6] strain was unique compared to the other five Brazilian G8P[6] strains described here. RVA/Human-wt/BRA/IAL-R2437/2010/G8P[6] displayed 100% aa homology in five VP7 gene antigenic regions (B–F) compared to the human RVA/Human-wt/CIV/6736/2004/G8P[8] and RVA/Human-wt/SVN/SI-885/2006/G8P[8] strains [53,54]. The alignment of aa sequences deduced from the VP7 gene revealed aa substitutions inside the variable region A (aa 39–50) at positions 43^I→V^, 44^V→I^ and 45^T→A^, region B (aa 87–101) at position 100^D→E^, region C (aa 120–130) at position 122^T/V→A^, region D (aa 143–152) at position 145^N→S^ and region F (aa 233–242) at position 237^V→I^. Amino acid substitutions were also observed outside VP7 hypervariable regions at positions 11^I→T^, 16^L→P^, 29^I→V^, 72^N/T→A^, 73^S/P→C^ and 268^V→I^. The VP7 protein of the RVA/Human-wt/BRA/IAL-R2437/2010/G8P[6] strain had two potential N-linked glycosylation sites located at aa 69-72 (NVSA) and 238-241 (NVTT) (Appendix A). 

Differences took place in the VP8* subunit variable region in the Brazilian G8P[6] DS-1-like strains at positions 135^R→K^ and 198^T→A^. Additionally, amino acid changes were found outside the VP4 hypervariable region in Brazilian G8P[6] DS-1-like strains at position 255^I→V^. In the VP4 hypervariable region, there was 100% aa homology between Brazilian G8P[6] DS-1-like strains and RVA/Human-wt/BRA/IAL-RN376/2009/G8P[6] strains previously reported in Brazil [26]. The six Brazilian G8P[6] DS-1-like strains preserved the potential cleavage sites of three arginines (R), 230, 240 and 246, maintaining the highly conserved cysteine (C) at residue 215 and prolines (P) at residues 68, 71, 224 and 225 (Appendix A).

Table 1 shows the six Brazilian G8P[6] DS-1-like lineage genotype constellations compared with those of other selected DS-1-like backbone RVA strains. Brazilian G8P[6] DS-1-like strains possess a unique lineage genotype constellation shared only with the RVA/Human-wt/COD/KisB554/2010/G8P[6] strain and represented by G8.V-P[6].Ia-A2.IVa-N2.V-T2.V-E2.XXI-H2.IVa-R2.VI-C2.IVa-M2.V-I1.V. The VP1-4, VP6-7 and NSP1-5/6 genes from the six Brazilian G8P[6] DS-1-like strains (IAL-R2437/2010, IAL-R2404/2010, IAL-RN377/2009, IAL-RN374/2009, IAL-RN373/2009 and IAL-RN361/2009) exhibited sequence nt identity ranging from 97.7 to 100% (Figure 1A,D–M). Phylogenetic analysis of the genes encoding neutralization antigens and the DS-1-like backbone together with representative RVA strains from around the world indicated that the Brazilian G8P[6] DS-1-like strains are genetically connected to African and American strains, more strongly with RVA/Human-wt/COD/KisB554/2010/G8P[6] and RVA/Human-wt/USA/06-242/2006/G2P[6] strains. Percentages of nt identity between Brazilian G8P[6] DS-1-like and the representative African and American RVA strains varied across the 11 gene segments: 99.1–100% for VP7, 99.2–100% for VP4, 99.3–99.9% for NSP1, 97.5–99.2% for NSP2, 99.2–99.5% for NSP3, 99.8–100% for NSP4, 99.8% for NSP5, 97.0–99.4% for VP1, 98.7–100% for VP2, 96.0–99.8% for VP3 and 99.5–99.9% for VP6 (Figure 1A,D–M).

The comparative VP7 sequence analysis revealed that the bovine RVA G8P[1] NGRBg8/1998 strain detected in Nigeria and the human RVA G8P[6] DS-1like strains reported in the present investigation share nucleotide identities varying from 95.6% to 95.9%, highlighting G8 strains’ potential ruminant ancestry (Figure 1A). In addition, genetic analysis of the VP4 gene revealed that the African straw-colored fruit bat (*Eidolon helvum*) RVA G25P[6] 4852/2007 strain detected in Kenya [22] also clustered inside Lineage I-a together with the six Brazilian human G8P[6] DS-1-like strains reported here, displaying 94.1–94.9% of nucleotide similarity among them (Figure 1D). 

### 3.5. Modeling of the VP7 Gene

In order to better understand the differences in amino acid composition and the antigenic characteristics of human and animal G8 strains, the amino acid sequences of the VP7 antigenic region were examined. The results were ranked according to the DOPE scores and graphed accordingly (Appendix A). 

Regarding the reference structures of human origin, the distance matrix shows similarities above 94% for the cases identified in this study. From the point of view of amino acids, similar values (91.46 up to 100%) were detected in the bovine samples as well. These similarities in themselves suggest that there is little difference in the antibody recognition of this protein in the human immune system. However, to verify this correctly, we made predictions regarding the interactions with antibodies and with discontinuous epitopes, which could emerge from the differences, however small they were. Structurally, the proteins diverged very little from the human and bovine references, as shown in Table 2, and structural alignment (Figure 2). Furthermore, the results for the detection of discontinuous epitopes and antibody interaction suggest overlap in all cases, thus excluding the possibility of differential epitopes based on observed differences.

## 4. Discussion

Sequencing data of Brazilian RVA G8 strains is very limited, especially considering full-genotyping or full-genome characterization. This extends to both human and animal strains [21,24,25,26,58]. Only four complete genomes of human RVA G8P[4] strains and one complete genome of bovine RVA G8P[11] strain detected in Brazil are available in the GenBank sequence database [21,58]. This study presents the full-genotype characterization of twelve RVA G8 strains, including the newly emergent bovine-like G8P[8] strain with the DS-1-like backbone, detected in distinct Brazilian regions. Whole-genotype characterization is crucial in tracking the emergence of novel RVA strains and understanding their evolution [59].

Strains exhibiting the G8 genotype are considered rare or uncommon [5,20]. The low frequency of RVA G8 infections detected in the present study (0.6%; 2007–2020) agreed with data previously described in Brazil in both pre- (1%; 2005) [27] and post-RVA vaccine eras (0.5%; 2007–2012) [47]. They were also similar to that observed in other studies carried out in Croatia (0.1%; 2012–2014) [60] and Thailand (0.6%; 2003–2004) [61]. A relatively high prevalence of G8 RVA strains has, for decades, been commonly observed in African countries [62,63,64,65,66]. Nevertheless, the increase in G8 detection outside Africa, such as in Asia, the Middle East and European countries [15,67,68] may imply that G8 strains are emerging across the globe and that this specific genotype should be carefully monitored. Oscillatory trends in the incidences of RVA genotypes are widely observed phenomena [3,5,29], and the emergence of the G8 could be explained by vaccine-induced genotypes and irregular RVA immunization schedules, or both [69,70]. Continued RVA surveillance is vital to better understand the contemporaneous role of G8 strains within human populations. In the present investigation, G8 strains followed their sporadic and confined pattern of detection in Brazil, thus not suggesting that a potential emergence is taking place in the country.

As is usual among RVA strains, the putative VP7 N-linked glycosylation site was located at amino acid (aa) 69 in Brazilian G8P[4]/P[6]/P[8] DS-1-like strains [71]. Additionally, like the majority of bovine and human G8 strains and Brazilian G8 strains reported here possessed a second glycosylation site at aa 238 [33,72]. Glycosylation of residue 238 has been observed to decrease the neutralization of animal G11 RVA strains by hyperimmune sera and MAbs, which may have broad implications for immunogenicity [73]. Inside the main antigenic site, region D (aa 143–152), an amino acid change at position 145^N→S^ in G8P[4] DS-1-like strains and at position 146^A→T^ in G8P[6] DS-1-like strains took place. Region E (aa 207–220), which is spatially close to region D, contains the amino acid substitution at position 218^I→V^ in G8P[4] DS-1-like strains and at position 218^V→I^ in G8P[6] DS-1-like strains [48]. The Brazilian G8P[8] DS-1-like strains did not exhibit amino acid substitutions in the D and E regions. Antigenic analyses using the VP7 gene did not reveal any distinctions between the epitopes of the G8 strains either from human or animal origin. 

The VP4 spike protein is cleaved by trypsin to produce the polypeptides VP8* and VP5*, which are needed to activate infectivity [74,75]. The Brazilian G8P[4]/P[6]/P[8] DS-1-like strains preserved the potential VP4 arginine cleavage sites (230, 240, and 246), assuring infectivity. The four proline residues (68, 71, 224 and 225) are also conserved. Given that proline is known to cause three-dimensional structural distortion, these conserved prolines may have a significant impact on the conformation of the VP4 [75]. A limitation of the current study was the failure to obtain entire sequences of the VP4 gene, impairing protein modeling analyses from being conducted. 

Atypical reassorted bovine-like G8P[8] strains with the DS-1-like backbone emerged during the 2013/2014 seasons in Southeast Asia [17,49,50,76], spreading to Europe and South America. They were also recently reported in the Czech Republic (2016–2018) [15] and Argentina (2018) [51,77]. Of the two Brazilian bovine-like G8P[8] strains with the DS-1-like backbone reported here, one (IAL-R193/2017) was acquired by a 7-month-old male patient in the city of Goiânia, Midwestern region. The other sample (IAL-R558/2017) was collected from a 4-month-old male child in the city of São Paulo, Southeastern region. These data indicate that the bovine-like G8P[8] DS-1-like strains have circulated in different Brazilian regions at the same time. Moreover, based on timeline detection data, it could be speculated that atypical bovine-like G8P[8] strains with the DS-1-like backbone reached the South America through Brazil (2017), then disseminated to other nations such as Argentina (2018) [51,77]. The route of bovine-like G8P[8] strains with the DS-1-like backbone spreading across the globe is tricky to be recognized, but globalization is probably the key point of RVA strain traffic from one continent to another [15].

An important issue to be highlighted is the fact that bovine-like G8P[8] strains with the DS-1-like backbone did not remain in circulation in the Brazilian population, since they were not detected in the subsequent years of the surveillance (2018 to 2020). This is different to what was observed with the equine-like G3P[8] DS-1-like strains [13]. Therefore, it can be suggested that bovine-like G8P[8] strains with the DS-1-like backbone do not achieve the fitness required to become a successful human pathogen in Brazil, as observed in Asian countries [17,50]. 

Phylogenetic analysis of the RVA/Human-wt/BRA/IAL-R193/2017/G8P[8] and RVA/Human-wt/BRA/IAL-R558/2017/G8P[8] VP7 gene segments showed that they clustered into G8 Lineage IV. The bovine strain from India (RVA/Cow-wt/IND/68/2007/G8P[14]) and human strains from Vietnam (RVA/Human-wt/VNM/RVN1149/2014/G8P[8]) and Thailand (RVA/Human-wt/THA/PCB-79/2013/G8P[8]; RVA/Human-wt/THA/SKT-457/2014/G8P[8]) are also included in Lineage IV. These human RVA bovine-like G8P[8] strains with the DS-1-like backbone that resemble cattle were identified in Asia between 2013 and 2014 and they confirmed the hypothesis of an interspecies transmission [17,50]. Our results indicate that the human G8P[8] strains with the DS-1-like backbone detected in this study are also bovine-like derived. 

On the one hand, the VP1 R2 genotype identified in the bovine-like G8P[8] strain with the DS-1-like backbone detected in Midwestern Brazil (IAL-R193/2017) belong to lineage XI and grouped together with the most recent bovine-like G8P[8] strains with the DS-1-like backbone detected since 2013 in Asia, including Japan, Vietnam, Thailand and Korea [17,49,50]. On the other hand, the VP1 R2 genotype recognized in the bovine-like G8P[8] strain with the DS-1-like backbone detected in Southeastern Brazil (IAL-R558/2017) possessed a distinct R2 lineage never previously described and grouped apart from any of the DS-1-like reference strains. Therefore, the bovine-like G8P[8] strains with the DS-1-like backbone detected in Midwestern and Southeastern Brazil in 2017 may have been introduced into the country from distinct pools of co-circulating bovine-like G8P[8] strains with the DS-1-like backbone. Intra-genotypic variability and distinct genotypic lineage constellations of the bovine-like G8P[8] strains with the DS-1-like backbone have been previously reported [78,79,80].

Additionally, the phylogenetic analysis of the NSP genes has demonstrated that the Brazilian bovine-like G8P[8] strains with the DS-1-like backbone clustered together with novel DS-1-like G1/G3/G9/G8P[8] strains detected in Asia, Europe and Americas [11,12,13,14,15,16,17], as well as with classical G2P[4] strains circulating in Australia in 1999 and in Japan in 2001 [43,81]. These findings collectively imply that the origin of the Brazilian bovine-like G8P[8] strains with the DS-1-like backbone is probably not directly related to importation from Asia, but rather that the atypical bovine-like G8P[8] strains with DS-1-like backbone continue to evolve, most likely through reassortment with regionally prevalent RVA strains. Over time, a globally co-circulating pool of different bovine-like G8P[8] strains with the DS-1-like backbone should be expected due to its natural evolution and/or rearrangements with local RVA strains. 

G8P[4] DS-1-like strains are found mainly in Africa (especially Malawi) and sporadically reported in Europe, Asia and the Americas, including in Brazil [21,25,52,62,63,82,83,84]. The four Brazilian G8P[4] DS-1-like strains revealed by phylogenetic research had nearly identical sequences when all 11 gene segments were taken into account. Additionally, a close relation was observed between Brazilian G8P[4] DS-1-like strains and two European G8P[4] strains for all genes investigated: the GER1H-09 isolated in Germany in 2009 [52] and the SS65 reported in Italy in 2011 [55]. Together, these findings suggested that the Brazilian G8P[4] DS-1-like strains were probably imported from Europe rather than being African-born. The VP7 gene of the Brazilian G8P[4] DS-1-like strains was the only genomic segment that, besides these two European G8P[4] strains, also clustered together with some Brazilian G8P[4] strains detected between 2010 and 2011 [21]. The origin of Brazilian G8P[4] DS-1-like strains described here may have involved reassortment events with locally G8 RVA circulating strains. 

All four Brazilian G8P[4] DS-1-like strains were identified in 2010 in the city of Brasilia (Brazil’s Capital), located within Goiás state (GO). Silva-Sales et al. [21] recently reported the full genome characterization of another four G8P[4] DS-1-like strains, also detected in 2010, but in the state of Tocantins (TO). Goiás and Tocantins are bordering states situated in the central region of the country, which is home to the Brazilian savanna biome. A significant finding could be drawn from this context. NSP1, NSP2 and NSP3 gene phylogenetic trees have shown that Brazilian Goiás G8P[4] DS-1-like strains and the Brazilian Tocantins G8P[4] DS-1-like strains, published previously [21], clustered in different lineages, suggesting genetic variety among Brazilian G8P[4] DS-1-like strains, which were essentially discovered at the same time and location. Collectively, the genomic analysis revealed that the Brazilian G8P[4] DS-1-like strains appeared to have undergone genetic reassortment events with both locally and globally circulating strains.

A potential interspecies transmission based on multiple reassortment events between artiodactyls, ruminant and human RVA strains were suggested for G8P[4] DS-1-like RVA strains detected in Asia [82,83]. The genetic analysis of the Brazilian DS-1-like G8P[4] RVA strains conducted here did not indicate a recent zoonotic origin, following previous phylogenetic investigations performed in European countries and Brazil [21,52,55]. Nevertheless, it is worth mentioning that a certain link between human and animal DS-1-like G8P[4] RVA strains does probably exist, as we recognized genetic relatedness of human DS-1-like G8P[4] VP7, NSP2 and NSP4 gene segments to those of bovine, sheep and dog RVA strains, respectively, attempting to hypothesize footprints of interspecies transmission events. More in-depth molecular analysis of DS-1-like G8P[4] RVA strains is hampered by a lack of genome sequencing data of RVA strains circulating in animals, and this is especially the case for Brazil. 

Significant epidemiological relevance has been placed on the G8P[6] DS-1-like genotype in Africa [57,62,63,85]. The Brazilian G8P[6] DS-1-like RVA strains reported here were detected from two different populations: four strains were obtained during an outbreak affecting Brazilian native children in the city of Dourados (Midwestern region) in 2009 [26] and two strains were acquired from children with acute gastroenteritis during the epidemiological survey in São Paulo city (Southeastern region) in 2010. The six G8P[6] DS-1-like strains were genetically similar to each other and clustered together in all 11 gene segments, therefore suggesting that, during those two years, the same G8P[6] DS-1-like strain was circulating throughout different parts of Brazil. 

The NSP3, NSP4, NSP5, VP1 and VP3 genes segments of the Brazilian G8P[6] DS-1-like strains clustered closely with human African RVA/Human-wt/COD/KisB554/2010/G8P[6] from the Democratic Republic of Congo [57]. Otherwise, NSP1, VP2 and VP6 genes were closely related to both African (RVA/Human-wt/COD/KisB554/2010/G8P[6]) and American (RVA/Human-wt/USA/06-242/2006/G2P[6]) strains [57]. The VP7 gene was related to African (RVA/Human-wt/COD/KisB554/2010/G8P[6]) and Argentinian G8P[6] strains [51,57]. Finally, VP4 and NSP2 genes segments grouped with G2P[4], G1P[6], G2P[6], G3P[6], G4P[6] and G8P[6] strains from Africa and USA [12,23,57]. All of these data appear to indicate that African genetic ancestry is present in Brazilian G8P[6] DS-1-like strains, although there is no doubt that these strains reassorted among nearby co-circulating American strains of the same DS-1 genotype constellation. Reassortment among co-circulating strains with the DS-1 genotype constellation is probably common, according to the phylogenetic analyses of Malawian G8P[6] DS-1-like strains conducted by Nakagomi et al. [85].

The Brazilian G8P[6] DS-1-like showed no evidence of recent zoonotic reassortment events, but genetic similarity between the African bat G25P[6] RVA strain and human G8P[6] RVA strains have been reported [22,26]. Genetic studies point to a possible porcine origin for the P[6] genotype [86]. The paucity of fresh conclusions drawn from the phylogenetic studies performed here is due to the dearth of information on animal P[6] strains. It is important to mention that four G8P[6] DS-1-like strains characterized here were characterized by Brazilian native children. It is well known that indigenous communities live in proximity to animals, sustaining the continuous exposure to potential interspecies transmission of RVA strains [26]. These data underscore the need for increased animal RVA molecular surveillance and attention to the value of a One Health strategy in the field of RVA research.

In conclusion, this is a pioneer study analyzing the complete constellation of G8P[4], G8P[6] and G8P[8] RVA strains detected in Brazil, as well as the first report of the novel bovine-like G8P[8] strains with the DS-1-like backbone circulating in the country. The genetic information obtained here has the potential to provide the basis for monitoring variations in the molecular composition of G8 RVA strains circulating in the Brazilian human population. Our findings highlight the variety of G8 RVA strains in Brazil and also contribute to the knowledge of G8P[4]/P[6]/P[8] RVA genetic diversity and evolution from a global perspective.

## Figures and Tables

**Figure 1 viruses-15-00664-f001:**
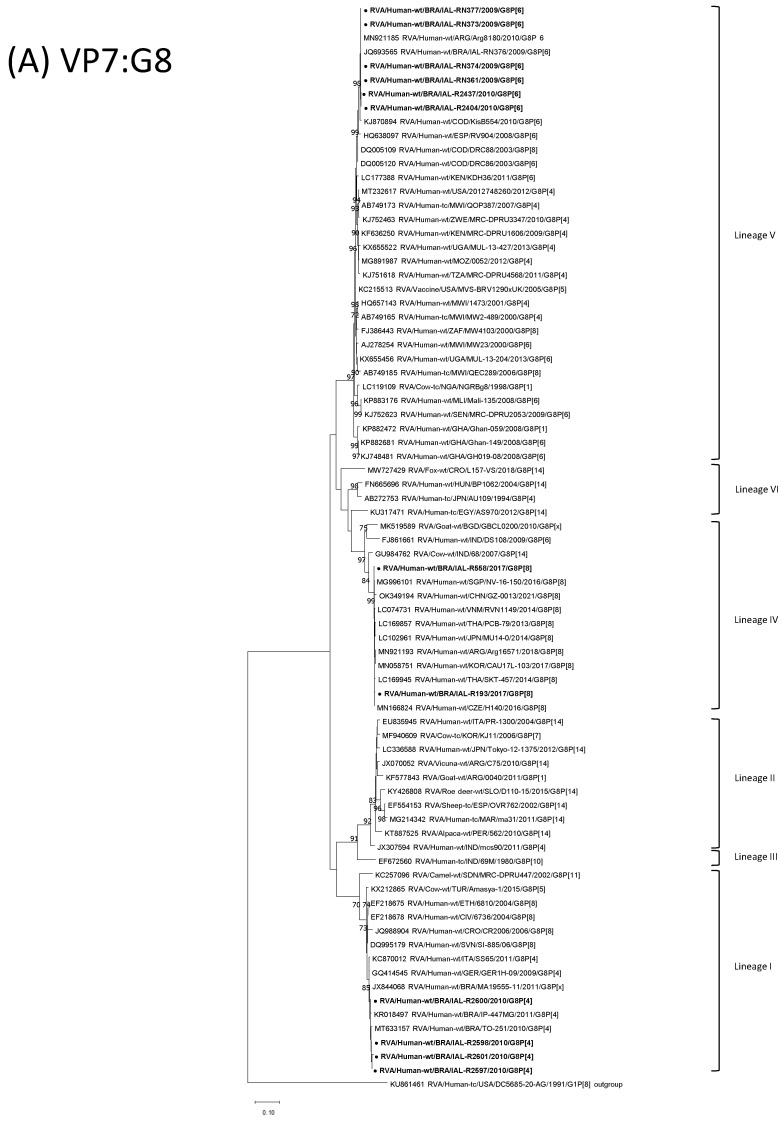
Nucleotide based phylogenetic relatedness of RVA/Human-wt/BRA/IAL-R193/2017/G8P[8], RVA/Human-wt/BRA/IAL-R558/2017/G8P[8], RVA/Human-wt/BRA/IAL-R2601/2010/G8P[4], RVA/Human-wt/BRA/IAL-R2600/2010/G8P[4], RVA/Human-wt/BRA/IAL-R2598/2010/G8P[4], RVA/Human-wt/BRA/IAL-R2597/2010/G8P[4], RVA/Human-wt/BRA/IAL-R2437/2010/G8P[6], RVA/Human-wt/BRA/IAL-R2404/2010/G8P[6], RVA/Human-wt/BRA/IAL-RN377/2009/G8P[6], RVA/Human-wt/BRA/IAL-RN374/2009/G8P[6], RVA/Human-wt/BRA/IAL-RN373/2009/G8P[6] and RVA/Human-wt/BRA/IAL-RN361/2009/G8P[6] strains (indicated in bold and by ●) (**A**) VP7:G8, (**B**) VP4:P[8], (**C**) VP4:P[4], (**D**) VP4:P[6], (**E**) VP1:R2, (**F**) VP2:C2, (**G**) VP3:M2, (**H**) VP6:I2, (**I**) NSP1:A2, (**J**) NSP2:N2, (**K**) NSP3:T2, (**L**) NSP4:E2 and (**M**) NSP5/6:H2 to other selected RVA strains. Maximum likelihood trees of complete or partial nucleotide sequences were generated with MEGA X software. Reference strains were obtained from the GenBank database. Genotypes, lineages, accession numbers, isolates, countries and the year of each strain are indicated. The scale indicates the number of divergent nucleotide residues. Percentages of bootstrap values are shown at the branch node.

**Figure 2 viruses-15-00664-f002:**
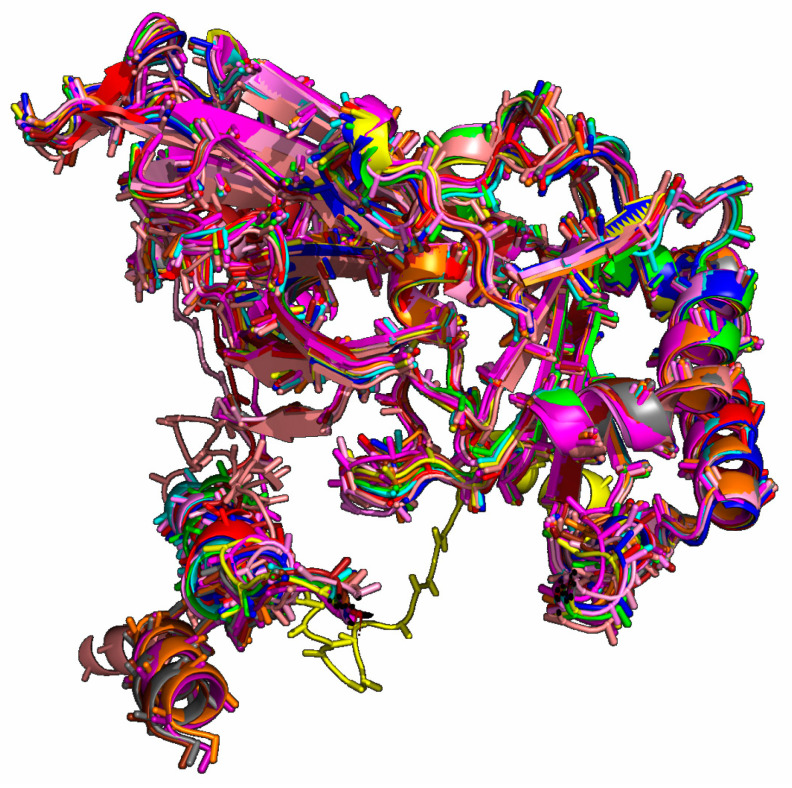
Structure alignment of outer capsid proteins VP7 G8 genotype of reference strains and selective Brazilian G8 strains (RVA/Human-wt/BRA/IAL-R193/2017/G8P[8], RVA/Human-wt/BRA/IAL-R558/2017/G8P[8], RVA/Human-wt/BRA/IAL-R2597/2010/G8P[4], RVA/Human-wt/BRA/IAL-R2598/2010/G8P[4] and RVA/Human-wt/BRA/IAL-R2601/2010/G8P[4]). Structures were evaluated according to the PDBSum GENERATE scores [46].

**Table 1 viruses-15-00664-t001:** Demographic and spatial data, migration profile, genotypes and lineage constellation of human rotavirus G8 strains, Brazil, 2007–2020. To aid visualization, lineage constellations of representative genotype 2 strains are highlighted in various colors.

Strain	Age	Gender	City	State	Profile	VP7	VP4			VP6	VP1	VP2	VP3	NSP1	NSP2	NSP3	NSP4	NSP5
						G8	P[8]	P[4]	P[6]	I2	R2	C2	M2	A2	N2	T2	E2	H2
RVA/Human-wt/BRA/IAL-R193/2017/G8P[8] ^a^	7 months	M	Goiânia	GO	Short	IV	III			V	XI	IVa	V	IVa	XV	V	XII	IVa
RVA/Human-wt/BRA/IAL-R558/2017/G8P[8] ^a^	4 months	M	São Paulo	SP	Short	IV	III			V	Distinct	IVa	V	IVa	XV	V	XII	IVa
RVA/Human-wt/BRA/IAL-R2601/2010/G8P[4] ^a^	1 year	F	Brasília	DF	Short	I		IVa		V	V	IVa	V	II	X	V	VII	IVa
RVA/Human-wt/BRA/IAL-R2600/2010/G8P[4] ^a^	5 years	F	Brasília	DF	Short	I		IVa		V	V	IVa	V	II	X	V	VII	IVa
RVA/Human-wt/BRA/IAL-R2598/2010/G8P[4] ^a^	5 months	F	Brasília	DF	Short	I		IVa		V	V	IVa	V	II	X	V	VII	IVa
RVA/Human-wt/BRA/IAL-R2597/2010/G8P[4] ^a^	5 years	M	Brasília	DF	Short	I		IVa		V	V	IVa	V	II	X	V	VII	IVa
RVA/Human-wt/BRA/IAL-R2437/2010/G8P[6] ^a^	9 months	M	São Paulo	SP	Short	V			I-a	V	VI	IVa	V	IVa	V	V	XXI	IVa
RVA/Human-wt/BRA/IAL-R2404/2010/G8P[6] ^a^	1 year	M	São Paulo	SP	Short	V			I-a	V	VI	IVa	V	IVa	V	V	XXI	IVa
RVA/Human-wt/BRA/IAL-RN377/2009/G8P[6] ^a^	3 months	F	Dourados	MS	Short	V			I-a	V	VI	IVa	V	IVa	V	V	XXI	IVa
RVA/Human-wt/BRA/IAL-RN374/2009/G8P[6] ^a^	5 months	M	Dourados	MS	Short	V			I-a	V	VI	IVa	V	IVa	V	V	XXI	IVa
RVA/Human-wt/BRA/IAL-RN373/2009/G8P[6] ^a^	2 months	F	Dourados	MS	Short	V			I-a	V	VI	IVa	V	IVa	V	V	XXI	IVa
RVA/Human-wt/BRA/IAL-RN361/2009/G8P[6] ^a^	1 year	M	Dourados	MS	Short	V			I-a	V	VI	IVa	V	IVa	V	V	XXI	IVa
**Bovine-like G8P[8] DS-1 like**																		
RVA/Human-wt/JPN/MU14-0/2014/G8P[8]						IV	III			V	XI	IVa	V	IVa	XV	V	XII	IVa
RVA/Human-wt/THA/SKT-457/2014/G8P[8]						IV	III			V	XI	IVa	V	IVa	XV	V	XII	IVa
RVA/Human-wt/THA/PCB-79/2013/G8P[8]						IV	III			V	XI	IVa	V	VII	XV	V	XII	IVa
RVA/Human-wt/VNM/RVN1149/2014/G8P[8]						IV	III			V	XI	IVa	V	IVa	XV	V	XII	IVa
RVA/Human-wt/KOR/CAU17L-103/2017/G8P[8]						IV	III			V	XI	IVa	V	IVa	V	V	XII	IVa
RVA/Human-wt/SGP/NV-16-150/2016/G8P[8]						IV	III			V	V	IVa	V	IVa	XV	V	XII	IVa
RVA/Human-wt/CHN/GZ-0013/2021/G8P[8]						IV	III			V	V	IVa	V	IVa	V	V	VI	IVa
**Equine-like G3P[4]**																		
RVA/Human-wt/JPN/S13-45/2013/G3P[4]								IVa		V	V	IVa	V	IVa	V	V	XI	IVa
**Equine-like G3P[8] reassortant**																		
RVA/Human-wt/AUS/D388/2013/G3P[8]							III			V	V	IVa	V	IVa	V	V	XI	IVa
RVA/Human-wt/BRA/IAL-R751/2016/G3P[8]							III			V	V	IVa	V	IVa	V	V	VI	IVa
RVA/Human-wt/DEU/GER33-15/2015/G3P[8]							III			V	V	IVa	V	IVa	V	V	VI	IVa
RVA/Human-wt/DEU/GER37-16/2016/G3P[8]							III			V	V	IVa	V	IVa	V	V	XI	IVa
RVA/Human-wt/HUN/ERN8148/2015/G3P[8]							III			V	V	IVa	V	IVa	V	V	VI	IVa
RVA/Human-wt/JPN/15R429/2015/G3P[8]							III			V	V	IVa	V	IVa	V	V	VI	IVa
RVA/Human-wt/ESP/SS98244047/2015/G3P[8]							III			V	V	IVa	V	IVa	V	V	VI	IVa
RVA/Human-wt/THA/SKT-289/2013/G3P[8]							III			V	V	IVa	V	IVa	V	V	XI	IVa
RVA/Human-wt/USA/3000390639/2015/G3P[8]							III			V	V	IVa	V	IVa	V	V	VI	IVa
**G1P[8] reassortant**																		
RVA/Human-wt/JPN/KN039/2012/G1P[8]							III			V	V	IVa	V	IVa	V	V	VI	IVa
RVA/Human-wt/JPN/KN041/2012/G1P[8]							III			V	V	IVa	V	II	V	V	VII	IVa
RVA/Human-wt/MWI/BID2AW/2013/G1P[8]							III			V	V	IVa	V	IVa	V	V	VII	IVa
RVA/Human-wt/MWI/BID1PU/2013/G1P[8]							III			V	V	IVa	V	IVa	V	V	XIII	IVa
RVA/Human-wt/PHI/TGO12-004/2012/G1P[8]							III			V	V	IVa	V	IVa	V	V	VI	IVa
RVA/Human-wt/THA/SKT-109/2013/G1P[8]							III			V	V	IVa	V	IVa	V	V	VI	IVa
RVA/Human-wt/VNM/SP026/2012/G1P[8]							III			V	V	IVa	V	IVa	V	V	VI	IVa
RVA/Human-wt/BRA/IAL-R3122/2013/G1P[8]							III			V	V	IVa	V	IVa	V	V	VI	IVa
**G9P[8] reassortant**																		
RVA/Human-wt/VNM/RVN16.1024/2016/G9P[8]							III			V	XI	IVa	V	IVa	V	V	XII	IVa
RVA/Human-wt/THA/DBM2017-203/2017/G9P[8]							III			V	V	IVa	V	IVa	V	V	VI	IVa
RVA/Human-wt/JPN/To14-37/2014/G9P[8]							III			V	XI	IVa	V	IVa	V	T1	E1	IVa
**G8P[4]**																		
RVA/Human-wt/MWI/1473/2001/G8P[4]						V		II		V	II	IVa	V	IVa	V	V	V	IVa
RVA/Human-tc/JPN/AU109/1994/G8P[4]						VI		IV non-a		IV	VIII	IV non-a	IV	IV non-a	VIII	Distinct	IV	IV non-a
RVA/Human-tc/MWI/MW2-489/2000/G8P[4]						V		II		V	II	IVa	V	IVa	V	V	V	IVa
RVA/Human-tc/MWI/QOP387/2007/G8P[4]						V		II		V	V	IVa	V	IVa	V	V	XXII	IVa
RVA/Human-wt/BRA/TO-251/2010/G8P[4]						I		IVa		V	V	IVa	V	IVa	V	V	V	IVa
RVA/Human-wt/UGA/MUL-13-427/2013/G8P4						V		II		V	V	IVa	V	IVa	V	V	XXIII	IVa
RVA/Human-wt/USA/2012748260/2012/G8P[4]						V		II		V	V	IVa	V	IVa	V	V	XXII	IVa
RVA/Human-wt/ZWE/MRC-DPRU3347/2010/G8P[4]						V		II		V	V	IVa	V	IVa	V	V	XXII	IVa
RVA/Human-wt/MOZ/0052/2012/G8P[4]						V		II		V	V	IVa	V	IVa	V	V	XXII	IVa
RVA/Human-wt/DEU/GER1H-09/2009/G8P[4]						I		IVa		V	V	IVa	V	II	X	V	VII	IVa
RVA/Human-wt/TZA/MRC-DPRU4568/2011/G8P[4]						V		II		V	V	IVa	V	IVa	V	V	XXII	IVa
RVA/Human-wt/KEN/MRC-DPRU1606/2009/G8P[4]						V		II		V	V	IVa	V	IVa	V	V	XXIII	IVa
RVA/Human-wt/ITA/SS65/2011/G8[4]						I		IVa		V	Rx	Cx	Mx	II	X	V	VII	IVa
RVA/Human-wt/IND/mcs90/2011/G8P[4]						II		IVa		V	V	IVa	VII	IVa	V	V	VI	H3
**G8P[6]**																		
RVA/Human-wt/GHA/Ghan-149/2008/G8P[6]						V			I-a	VI	VI	VI	XIII	IVa	VII	V	XXI	H3
RVA/Human-wt/MLI/Mali-135/2008/G8P[6]						V			I-a	IX	VI	IVa	VI	IVa	XI	V	Distinct	IVa
RVA/Human-wt/ COD/KisB554/2010/G8P[6]						V			I-a	V	VI	IVa	V	IVa	V	V	XXI	IVa
RVA/Human-wt/COD/DRC86/2003/G8P[6]						V			I-a	V	V	IVa	V	IVa	V	V	XV	IVa
RVA/Human-wt/UGA/MUL-13-204/2013/G8P[6]						V			I-a	VI	V	IVa	V	IVa	V	V	XXXI	IVa
RVA/Human-wt/SEN/MRC-DPRU2053/2009/G8P[6]						V			I-a	IX	VI	IVa	VI	IVa	XI	V	Distinct	IVa
RVA/Human-wt/IND/DS108/2009/G8P[6]						IV			I-a	V	Rx	Cx	Mx	IVa	V	V	E9	IVa
RVA/Human-wt/GHA/GH019-08/2008/G8P[6]						V			I-a	VI	V	VI	XIII	IVa	VII	V	XXI	H3
**G8P[8] DS-1 like**																		
RVA/Human-wt/COD/DRC88/2003/G8P[8]						V	III			V	V	IVa	V	IVa	V	V	XV	IVa
RVA/Human-tc/MWI/QEC289/2006/G8P[8]						V	III			V	V	IVa	V	IVa	V	V	V	IVa
**G4P[6]**																		
RVA/Human-wt/ZMB/MRC-DPRU1752/XXXX/G4P[6]									I-a	V	V	IVa	V	IVa	V	V	VII	IVa
**G8P[11]**																		
RVA/Camel-wt/SDN/MRC-DPRU447/2002/G8P[11]						I				X	V	XII	XII	Ax	IX	T6	XIX	H3
**Vaccines strains**																		
RVA/Vaccine/USA/Rotateq-WI79-4/1992/G6P[8]							II			X	XII	X	X	A3	XIII	T6	XXIX	H3
**G8P[14]**																		
RVA/Human-wt/JPN/TOKYO/12-1375/2012/G8P[14]						II				X	XII	X	X	A3	XIII	T6	XXIV	H3
RVA/Human-wt/HUN/BP1062/2004/G8P[14]						VI				VIII	VIII	VII	VI	A11	Distinct	T6	XX	H3
RVA/Vicuna-wt/ARG/C75/2010/G8P[14]						II				Distinct	XII	XII	XV	Ax	XIX	T6	E3	Hx
RVA/Sheep-tc/ESP/OVR762/2002/G8P[14]						II				IX	IX	XI	VII	A11	X	T6	XX	H3
RVA/Human-tc/EGY/AS970/2012/G8P[14]						VI				XIV	X	IX	Distinct	A11	X	T6	VII	H3
RVA/Human-tc/MAR/ma31/2011/G8P[14]						II				IX	Distinct	IX	VIII	A11	X	T6	XX	H3
RVA/Alpaca-wt/PER/562/2010/G8P[14]						II				Distinct	Distinct	-	VIII	Ax	XV	T6	E3	H3
RVA/Roe deer-wt/SLO/D110-15/2015/G8P[14]						II				XIII	IX	IX	X	A3	XV	T6	XVIII	H3
**G8P[1]**																		
RVA/Human-wt/GHA/Ghan-059/2008/G8P[1]						V				VI	VI	IX	VI	A11	IX	T6	IX	H3
RVA/Goat-wt/ARG/0040/2011/G8P[1]						II				XIII	Distinct	-	XV	A3	XV	T6	E12	H3
RVA/Cow-tc/NGA/NGRBg8/1998/G8P[1]						V				XIV	VI	X	VIII	A11	VII	T6	IX	H3
**G2P[6]**																		
RVA/Human-wt/MLI/Mali-028/2008/G2P[6]									I-a	V	V	IVa	V	IVa	XV	V	Distinct	IVa
RVA/Human-wt/MWI/BID15I/2012/G2P[6]									I-a	V	V	IVa	V	IVa	V	V	VII	IVa
RVA/Human-wt/GHA/Ghan-108/2009/G2P[6]									I-a	V	V	IVa	V	IVa	V	V	X	IVa
RVA/Human-wt/USA/06-242/2006/G2P[6]									I-a	V	V	IVa	V	IVa	V	V	Distinct	IVa
**G3P[6]**																		
RVA/Human-wt/USA/3000354444/2015/G3P[6]									I-a	V	VI	IVa	V	IVa	V	V	IX	IVa
RVA/Human-wt/GHA/Ghan-105/2009/G3P[6]									I-a	V	V	IVa	V	IVa	N1	V	Distinct	H3
**G6P[6]**																		
RVA/Human-wt/BEL/B1711/2002/G6P[6]									I-a	V	VI	IVa	XI	IVa	V	V	V	IVa
RVA/Human-wt/CMR/ES298/2011/G6P[6]									I-a	V	V	IVa	V	IVa	V	V	XV	IVa
RVA/Human-wt/ITA/CEC06/2011/G6P[6]									I-a	VI	Rx	Cx	Mx	IVa	V	V	XV	IVa
**G9P[4]**																		
RVA/Human-wt/IND/RV09/2009/G9P[4]								IVa		V	V	IVa	V	IVa	V	T1	VI	IVa
RVA/Human-wt/IND/RV10/2010/G9P[4]								IVa		V	V	IVa	VII	IVa	V	V	E6	IVa
RVA/Human-wt/IND/kol-047/2013/G9P[4]								IVa		V	V	IVa	VII	IVa	N1	V	E6	IVa
RVA/Human-wt/ITA/AN19/2016/G9P[4]								IVa		V	V	IVa	VII	IVa	V	V	VI	IVa
RVA/Human-wt/JPN/S120088/2012/G9P[4]								IVa		V	V	IVa	V	IVa	V	T1	V	IVa
RVA/Human-wt/USA/LB1562/2010/G9P[4]								IVa		V	V	IVa	VII	IVa	V	V	E6	IVa
**G2P[4]**																		
RVA/Human-wt/JPN/01P007/2001/G2P[4]								IVa		V	V	IVa	V	IVa	V	V	V	IVa
RVA/Human-wt/USA/VU10-11-9/2011/G2P[4]								IVa		V	V	IVa	V	IVa	V	V	VI	IVa
RVA/Human-wt/USA/2007769964/2007/G2P[4]								IVa		V	V	IVa	V	II	V	V	VII	IVa
RVA/Human-wt/MLW/BID1JK/2013/G2P[4]								IVa		V	V	IVa	V	IVa	V	V	XIII	IVa
RVA/Human-wt/VNM/NT0578/2008/G2P[4]								IVa		V	V	IVa	V	IVa	VIII	V	VIII	IVa
RVA/Human-wt/TGO/MRC-DPRU5124/2010/G2P[4]								IVa		V	V	IVa	V	IVa	V	V	IX	IVa
RVA/Human-wt/GHA/GHNAV483/2009/G2P[4]								IVa		V	V	IVa	V	IVa	V	V	X	IVa
RVA/Human-wt/MWI/BID1CT/2012/G2P[4]								IVa		V	V	IVa	V	IVa	V	V	XIII	IVa
RVA/Human-wt/GHA/GHPML1989/2012/G2P[4]								IVa		V	V	IVa	VII	IVa	V	V	VI	IVa
RVA/Human-wt/CHN/TB-Chen/1996/G2P[4]								IV non-a		IV	IV	IV non-a	IV	IV non-a	IV	Distinct	IV	IV non-a
RVA/Human-wt/JPN/KUN/1980/G2P[4]								III		III	III	III	III	III	III	III	III	III
RVA/Human-wt/ITA/PAI11/1996/G2P[4]								II		II	II	II	II	II	II	II	II	II
RVA/Human-wt/USA/DS-1/1976/G2P[4]								I		I	I	I	I	I	I	I	I	I
RVA/Human-wt/BRA/TO-095/2015/G2P[4]								IVa		V	V	IVa	V	IVa	V	V	V	IVa
RVA/Human-tc/JPN/AU605/1986/G2P[4]								IV non-a		IV	IV	IV non-a	IV	IV non-a	N1	IV	IV	IV non-a
RVA/Human-wt/JPN/89Y1520/1989/G2P[4]								IV non-a		IV	IV	IV non-a	IV	IV non-a	N1	IV	IV	IVa
RVA/Human-wt/BGD/MMC88/2005/G2P[4]								IVa		V	V	IVa	VII	IVa	V	V	V	IVa
RVA/Human-wt/PRY/1040SR/2005/G2P[4]								IVa		V	V	IVa	V	IVa	V	V	V	IVa
RVA/Human-wt/ITA/PA150/2006/G2P[4]								IVa		V	V	IVa	V	IVa	V	V	V	IVa
RVA/Human-wt/HUN/ERN5603/2012/G2P[4]								IVa		V	V	IVa	V	IVa	V	V	VII	IVa
RVA/Human-wt/AUS/V233/1999/G2P[4]								IVa		V	V	IVa	V	IVa	V	V	V	IVa
RVA/Human-wt/CAN/RT036-07/2007/G2P[4]								IVa		V	V	IVa	V	IVa	V	V	VII	IVa
RVA/Human-wt/MWI/BID11S/2012/G2P[4]								IVa		V	V	IVa	V	IVa	V	V	VII	IVa

GO: Goiás state; SP: São Paulo state; DF: Federal District (Brazil Capital); MS: Mato Grosso do Sul state. ^a^ Strains characterized in the presente study.

**Table 2 viruses-15-00664-t002:** *Root Mean Square Deviation* (*RMSD*) deviations values of VP7 gene G8 genotype of reference strains and selective Brazilian G8 strains (RVA/Human-wt/BRA/IAL-R193/2017/G8P[8], RVA/Human-wt/BRA/IAL-R558/2017/G8P[8], RVA/Human-wt/BRA/IAL-R2597/2010/G8P[4], RVA/Human-wt/BRA/IAL-R2598/2010/G8P[4] and RVA/Human-wt/BRA/IAL-R2601/2010/G8P[4]).

Protein	GQ225781 Bovine Chain_A_A	GU984760 Bovine Chain_A_A	KF305321 Bovine Chain_A_A	KX212865 Bovine Chain_A_A	LC119109 Bovine Chain_A_A	MF940609 Bovine Chain_A_A	MT633156 Human Chain_A_A	MN989610 Human Chain_A_A	LC102961 Human Chain_A_A	IAL-R193 Human Chain_A_A	IAL-R558 Human Chain_A_A	IAL-R2597 Human Chain_A_A	IAL-R2598 Human Chain_A_A	IAL-R2601 Human Chain_A_A
**GQ225781**	None	0.8403	0.71	1.1921	0.6988	0.7066	2.2261	1.5824	1.6236	1.2486	1.2736	1.322	1.2609	1.2433
**GU984760**	0.8403	None	0.7676	1.0677	0.756	0.7377	2.3079	1.5151	1.6768	1.1326	1.0816	1.2795	1.0754	1.0572
**KF305321**	0.71	0.7676	None	1.1332	0.7233	0.7164	2.2884	1.5478	1.6369	1.2352	1.2762	1.3485	1.2323	1.2301
**KX212865**	1.1921	1.0677	1.1332	None	1.1732	1.1352	1.9266	1.2737	1.1325	0.999	0.9146	1.0105	0.9257	0.931
**LC119109**	0.6988	0.756	0.7233	1.1732	None	0.6704	2.2777	1.6554	1.7588	1.1886	1.2276	1.4169	1.2077	1.1962
**MF940609**	0.7066	0.7377	0.7164	1.1352	0.6704	None	2.2617	1.5988	1.725	1.1507	1.2141	1.3857	1.1806	1.1678
**MT633156**	2.2261	2.3079	2.2884	1.9266	2.2777	2.2617	None	2.4741	2.8254	2.0582	2.0242	2.4111	2.0232	2.0604
**MN989610**	1.5824	1.5151	1.5478	1.2737	1.6554	1.5988	2.4741	None	2.2828	1.2089	1.2356	1.6631	1.1921	1.2006
**LC102961**	1.6236	1.6768	1.6369	1.1325	1.7588	1.725	2.8254	2.2828	None	0.947	1.0429	2.4073	1.0142	1.021
**IAL-R193**	1.2486	1.1326	1.2352	0.999	1.1886	1.1507	2.0582	1.2089	0.947	None	0.5722	0.9145	0.5865	0.5743
**IAL-R558**	1.2736	1.0816	1.2762	0.9146	1.2276	1.2141	2.0242	1.2356	1.0429	0.5722	None	0.9316	0.4324	0.4272
**IAL-R2597**	1.322	1.2795	1.3485	1.0105	1.4169	1.3857	2.4111	1.6631	2.4073	0.9145	0.9316	None	0.8516	0.8224
**IAL-R2598**	1.2609	1.0754	1.2323	0.9257	1.2077	1.1806	2.0232	1.1921	1.0142	0.5865	0.4324	0.8516	None	0.2972
**IAL-R2601**	1.2433	1.0572	1.2301	0.931	1.1962	1.1678	2.0604	1.2006	1.021	0.5743	0.4272	0.4324	0.2972	None

G8 genotype VP7 gene structures were aligned and the root mean square deviations were calculated for the Ca of each aligned structure against each other. Reference strains were obtained from GenBank database. Accession numbers of the reference strains and Brazilian G8 strains reported in the present study are indicated. RMSD calculations were conducted using the PyMod modeler module (SAlign) from the Pymol 2.5 (https://pymol.org/2/).

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
