# Peer review of "Genomic Constellation of Human Rotavirus G8 Strains in Brazil over a 13-Year Period: Detection of the Novel Bovine-like G8P[8] Strains with the DS-1-like Backbone"

_viruses, 2023, doi:10.3390/v15030664_

Round 1

Reviewer 1 Report

The manuscript entitled “Genomic constellation of human rotavirus G8 strains in Brazil 2 over a 13-year period: detection of the novel bovine-like G8P[8] 3 DS-1-like and footprints of interspecies transmission” presented the complete genotype characterization of twelve RNA G8 strains detected in Brazil from year 2009 to year 2020, including the newly emerging bovine-like G8P[8] DS-1-like strain. The phylogenetic analysis reinforced for the evidence for bovine-to-human G8 RVA transmissions and the genetic information is helpful for monitoring variations of G8 RVA strains. However, there are several issues need to be addressed before considered for publication:

 1.      This paper only provided the phylogenetic analysis of the strains, which did not involving the analysis of those mutation sites. I suggests the author should provide the results of multiple sequence alignment and point out the mutation sites between the detected strains from Brazil and reference strain or previously transmitted strains. The mutation sites on both structure and non-structure protein need to be analyzed.

2.      The mutation sites on the structure protein need to be mapped onto the 3-D structures, and compare its location with the reported epitopes or binding sites to illustrate whether those mutations could influence the antigenicity or the capability for the virus to infect human. If the author want to discuss the possibility of bovine-to-human transmission, they need to provide the comparison of bovine related binding sites and human related binding sites to see the differences.

3.      The outer capsid proteins VP7 (capsid glycoprotein) and VP4 (spike protein) independently elicit neutralizing antibodies and form the basis of binary classification system of G and P types, respectively. The author should shown that whether the phylogenetic tree analysis of other non-structure proteins also follows the clustering of G or P types, otherwise, recombination events might be involved in those strains.

4.      Also, I suggests the authors to provide the recombination analysis of those detected strains to see if the circulating strains in Brazil involving the genomes from Bovine. This might be an important evidence for bovine-to-human transmission.

 5.      There are several typos in the manuscript, for example, in P2 L93: a total of 12.978 stool samples-> a total of 12,978 stool samples. This kind of typos may totally change the meaning of the sentence, which need to be carefully checked.

Author Response

The manuscript entitled “Genomic constellation of human rotavirus G8 strains in Brazil over a 13-year period: detection of the novel bovine-like G8P[8] DS-1-like and footprints of interspecies transmission” presented the complete genotype characterization of twelve RNA G8 strains detected in Brazil from year 2009 to year 2020, including the newly emerging bovine-like G8P[8] DS-1-like strain. The phylogenetic analysis reinforced for the evidence for bovine-to-human G8 RVA transmissions and the genetic information is helpful for monitoring variations of G8 RVA strains. However, there are several issues need to be addressed before considered for publication:

  1. This paper only provided the phylogenetic analysis of the strains, which did not involving the analysis of those mutation sites. I suggests the author should provide the results of multiple sequence alignment and point out the mutation sites between the detected strains from Brazil and reference strain or previously transmitted strains. The mutation sites on both structure and non-structure protein need to be analyzed.

Thank you for your suggestion. Hypervariable regions of VP7 and VP4 gene segments were analyzed. Please, see methods, results and discussion topics highlighted in yellow, together with supplementary material uploaded.

  1. The mutation sites on the structure protein need to be mapped onto the 3-D structures, and compare its location with the reported epitopes or binding sites to illustrate whether those mutations could influence the antigenicity or the capability for the virus to infect human. If the author want to discuss the possibility of bovine-to-human transmission, they need to provide the comparison of bovine related binding sites and human related binding sites to see the differences.

We appreciate your suggestion. VP7 protein modeling was included in the study. Please, see methods and results topics highlighted in yellow, together with figures, tables and supplementary material uploaded.

  1. The outer capsid proteins VP7 (capsid glycoprotein) and VP4 (spike protein) independently elicit neutralizing antibodies and form the basis of binary classification system of G and P types, respectively. The author should shown that whether the phylogenetic tree analysis of other non-structure proteins also follows the clustering of G or P types, otherwise, recombination events might be involved in those strains.

The clustering pattern of all VPs and NSPs gene segments of each strain was analyzed and discussed throughout the original manuscript.

  1. Also, I suggests the authors to provide the recombination analysis of those detected strains to see if the circulating strains in Brazil involving the genomes from Bovine. This might be an important evidence for bovine-to-human transmission.

Homologous recombination is thought to be especially rare in rotaviruses due to their segmented dsRNA genomes and their polymerase’s transcription and replication mechanisms. dsRNA viruses cannot easily undergo intragenic recombination because their genomes are not replicated in the cytoplasm by host polymerases. The most important mechanism in rotavirus evolution is reassortment events, whereby the 11 double-stranded RNA genome segments are exchanged among strains during co-infection. This mechanism directly affects the clustering pattern of the 11 gene segments of the rotavirus strains, phenom extensively analyzed in the phylogenetic trees and stressed out in results and discussion topics.

  1. There are several typos in the manuscript, for example, in P2 L93: a total of 12.978 stool samples-> a total of 12,978 stool samples. This kind of typos may totally change the meaning of the sentence, which need to be carefully checked.

Response to question 5. Typos were checked.

Reviewer 2 Report

Title: Genomic constellation of human rotavirus G8 strain in Brazil over a 13-year period’ detection of the novel bovine-like G8P]8\ DS-1-like and footprints of interspecies transmission.

Authors: Medeiros, R.S., et al.

This study determined the genotype constellation of 12 G8 strains from convenient surveillance specimens collected in Brazil between 2009 and 2017, and found that, while all of them had the DS-1 backbone genes, namely, I2-R2-C2-M2-A2-N2-T2-E2-H2, six samples from the 2009-2010 collection had G8P[6], four from the 2010 collection had G8P[4], and the most recent two samples collected in 2017 had “bovine-like G8P[8]” which was very similar to those emerged first in South-East Asian countries and then spread more globally. The authors conducted phylogenetic analysis on each genome segment of these 12 G8 strains and described what strains were most closely related to theirs among those available in the GenBank database. They also found some trait of zoonotic origin in such genes as the NSP2 and NSP4 genes of the G8P[4] strains. The paper contains potentially interesting observations that will contribute to a better understanding of the molecular epidemiology and evolution of rare, yet globally detectable G8 strains in humans. However, there are a few issues that need to be clarified or modified to conform to the existing literature. Thus, this reviewer suggests the authors that the manuscript be revised in accordance with the following comments.

Major comments:

Firstly, Some of the sequence data the authors generated in this study are much shorter than they are called “nearly full-length”, to say nothing of covering the entire coding sequence. Take the VP1 sequences as an example, what they determined was as short as around 600 nt out of the entire sequence of 3300 nt. In this regard, the notion that “Complete or nearly complete nucleotide sequences for 11 genome segments of twelve selected G8 strains were determined” (lines 170-171) is not supported by the data provided to this reviewer, and unacceptable.

While the authors may argue that the designation of the genotype or lineage of such genes was unaffected by the brevity of the sequence they determined, the point that this reviewer wishes to make is that the sequence information itself has fundamental importance in the original scientific paper like this one, not to give the reader a false impression. So, please provide as a supplementary material the table which shows the length of the sequence determined including from what nucleotide position to what nucleotide position for all genome segments. In addition, it should be stated in the first part of the Result section what percentages (showing by range) of the genome segments were determined and that “The length of the sequences we determined for the twelve G8 strains and the nucleotide positions compared are shown in supplementary Table 1”.

Secondly, G8 strains detected in humans are well known to have the DS-1 like genotype constellation; i.e., I2-R2-C2-M2-A2-N2-T2-E2-H2. So, Table 1 provides little help in comparing the genetic composition of the internal capsid and non-structural protein genes within the authors’ own strains or between theirs and similar strains previously published in the literature. This reviewer strongly recommends the authors to provide a revised Table 1 in which to show the genotype/lineage constellations of their own strains as well as those of relevant reference strains. This should be feasible since the authors have already adopted the lineage designation scheme proposed by Agbemabiese, et al. (1) in the phylogenetic trees (Figure 1). Based on Figure 1 and based on this reviewer’s own analysis using the fasta file provided by the authors for this review, there are FOUR genotype/lineage constellations in their collections, which more properly represent the diversity of G8 strains that circulated in Brazil than the three genotype constellations that co-segregated with three P genotypes; i.e., G8P[4], G8P[6], and G8P[8]. The four genotype constellations are G8-P[8]-I2.V-R2.XI-C2.IVa-M2.V-A2.IVa-N2.XV-T2.V-E2.XII-H2.IVa (R193), G8-P[8]-I2.V-R2.distinct-C2.IVa-M2.V-A2.IVa-N2.XV-T2.V-E2.XII-H2.IVa (R558), G8-P[4]-I2.V-R2.V-C2.IVa-M2.V-A2.II-N2.X-T2.V-E2.VII-H2IVa (R2601, R2600, R2598, and R2597), and G8-P[6]- I2.V-R2.VI-C2.IVa-M2.V-A2.IVa-N2.V-T2.V-E2.XXI-H2.IVa (R2437, 2404, RN377, RN374, RN373, and RN361). As these sequences of letters are hard to recognise at a glance, colour-coding may be in order like Fig. 1 of Agbemabiese, et al.’s paper (1) or Table 1 of Hoa-Tran, et al.’s paper (Hoa-Tran TN, et al. Detection of three independently-generated DS-1-like G9P[8] reassortant rotavirus A strains during the G9P[8] dominance in Vietnam, 2016-2018. Infect Genet Evol 2020;80:104194). Also, almost invisible Figure 1 should be replaced by the revised Table 1. If the authors wish to keep Figure 1, only relevant lineages should be shown by which this reviewer means “sub-tree” so that the reader can read the strain names. However, what this reviewer recommend is to transfer the entire Figure 1 to supplementary materials.

            The discussion section should be reorganised in reference to the revised Table 1. Rather than what strain is most closely related to each of their G8P[4], G8P[6] and G8P[8] strains in each genome segment, similarity or distinctness of genotype/lineage constellations of four G8 strains detected in Brazil should be discussed in reference to those of G8 strains or other G types possessing the DS-1 backbone genes. From the genotype/lineage constellations of four G8 strains, it is immediately clear that while R193 has an identical genotype/lineage constellation of the DS-1-like, bovine-like G8P[8] strains possessing the DS-1 backbone genes that emerged and spread in Vietnam (such as RVN14.1149), R558 has a unique constellation in which the VP1 gene was reassorted with an yet undescribed, distinct lineage on the backbone lineage constellation identical to that of R193 and other bovine-like G8P[8] strains. This point is completely dropped from the original manuscript of the authors. Even worse is that the nature of this unique sequence is stated erroneously to be “similar to the German equine-like DS-1-like G3P[8] GER33-15 strain [63]” (lines 411-412). When this reviewer constructed a phylogenetic tree of the VP1 genes, the VP1 gene of GER33-15 was grouped clearly into lineage V just like that of R2601, and never close to that of R558. By the way the nt identity between GER33-15 and R558 was, according to this reviewer’s calculation, 92.9%, which, the authors may claim, was “the closest”, but such statement is nothing but misleading.

            In general, as to the interpretation of the phylogenetic trees, what the authors are stating is more like a repetition of what has already documented by the previous investigators (such as the artiodactyl origin of NSP2 and NSP4 genes of bovine-like G8P[8] strains), and it seems to this reviewer that the authors did not try to get any new insight (such as the one this reviewer pointed out above) that their new sequence data may potentially contain.

Thirdly, what surprises this reviewer is lack of reference to the work done by the Liverpool investigators who discovered the abundance of G8 strains in Africa for the first time as early as around the turn of the last century (Cunliffe NA, et al. Rotavirus G and P types in children with acute diarrhea in Blantyre, Malawi, from 1997 to 1998: predominance of novel P[6]G8 strains. J Med Virol. 1999; 57:308-312, Cunliffe NA, et al. Molecular and serologic characterization of novel serotype G8 human rotavirus strains detected in Blantyre, Malawi. Virology. 2000;274:309-320), and published about a decade ago a monumental paper similar, regarding what was done, to the authors’ current manuscript (Nakagomi T, et al. G8 rotaviruses with conserved genotype constellations detected in Malawi over 10 years (1997-2007) display frequent gene reassortment among strains co-circulating in humans. J Gen Virol 2013; 94:1273-1295). In the latter paper, it was concluded that “Malawian G8 strains are closely related to other human strains with the DS-1 genotype constellation. They have evolved over the last decade through genetic reassortment with other human rotaviruses, changing their VP4 genotypes while maintaining a conserved genotype constellation for the remaining structural and non-structural proteins.” Do the authors not challenge this earlier observation in Africa, do they? Were the previous investigators wrong or was the difference attributable to the difference in the way the G8 viruses are circulating in the two continents? This reviewer believe that the authors would make a insightful interpretation in the revised manuscript.

Minor comments:

The sequences that are provided to this reviewer and the ones that the authors used to draw the phylogenetic trees in Figure 1 seem different. Check if the VP7 sequence of R2437 is correct.

Figure 1 (E) VP1:R2, The lineage that R2061, R2598, R2600 and R2597 belong to should be labelled as “Lineage V”

Figure 1 (I) NSP1:A2, The lineage that R193, R558, R2404, R2437, RN377, RN374, RN373, and RN361 belong to should be labelled as “Lineage IVa”

Author Response

Major comments:

Firstly, Some of the sequence data the authors generated in this study are much shorter than they are called “nearly full-length”, to say nothing of covering the entire coding sequence. Take the VP1 sequences as an example, what they determined was as short as around 600 nt out of the entire sequence of 3300 nt. In this regard, the notion that “Complete or nearly complete nucleotide sequences for 11 genome segments of twelve selected G8 strains were determined” (lines 170-171) is not supported by the data provided to this reviewer, and unacceptable.

Response to major comment 1. Thank you for your observation. As the reviewer should have noted we were very cautious in reinforce in the entire manuscript that we focused on full-genotyping characterization and not full-genome. We did not have the resources to obtain full genome sequence for the large segments (i.e., VP1-4), but we did obtain full genome sequence or nearly full-length for all NSPs gene segments and VP7 and VP6. That was the intention when writing the sentence. We rewrite the sentence in order to clarify the accomplished, and the entire manuscript was revised to avoid potential misunderstandings.

While the authors may argue that the designation of the genotype or lineage of such genes was unaffected by the brevity of the sequence they determined, the point that this reviewer wishes to make is that the sequence information itself has fundamental importance in the original scientific paper like this one, not to give the reader a false impression. So, please provide as a supplementary material the table which shows the length of the sequence determined including from what nucleotide position to what nucleotide position for all genome segments. In addition, it should be stated in the first part of the Result section what percentages (showing by range) of the genome segments were determined and that “The length of the sequences we determined for the twelve G8 strains and the nucleotide positions compared are shown in supplementary Table 1”.

Response to major comment 2. We included the clarification suggested in Results topic and provided the supplementary material (supplement 1).

Secondly, G8 strains detected in humans are well known to have the DS-1 like genotype constellation; i.e., I2-R2-C2-M2-A2-N2-T2-E2-H2. So, Table 1 provides little help in comparing the genetic composition of the internal capsid and non-structural protein genes within the authors’ own strains or between theirs and similar strains previously published in the literature. This reviewer strongly recommends the authors to provide a revised Table 1 in which to show the genotype/lineage constellations of their own strains as well as those of relevant reference strains. This should be feasible since the authors have already adopted the lineage designation scheme proposed by Agbemabiese, et al. (1) in the phylogenetic trees (Figure 1). Based on Figure 1 and based on this reviewer’s own analysis using the fasta file provided by the authors for this review, there are FOUR genotype/lineage constellations in their collections, which more properly represent the diversity of G8 strains that circulated in Brazil than the three genotype constellations that co-segregated with three P genotypes; i.e., G8P[4], G8P[6], and G8P[8]. The four genotype constellations are G8-P[8]-I2.V-R2.XI-C2.IVa-M2.V-A2.IVa-N2.XV-T2.V-E2.XII-H2.IVa (R193), G8-P[8]-I2.V-R2.distinct-C2.IVa-M2.V-A2.IVa-N2.XV-T2.V-E2.XII-H2.IVa (R558), G8-P[4]-I2.V-R2.V-C2.IVa-M2.V-A2.II-N2.X-T2.V-E2.VII-H2IVa (R2601, R2600, R2598, and R2597), and G8-P[6]- I2.V-R2.VI-C2.IVa-M2.V-A2.IVa-N2.V-T2.V-E2.XXI-H2.IVa (R2437, 2404, RN377, RN374, RN373, and RN361). As these sequences of letters are hard to recognise at a glance, colour-coding may be in order like Fig. 1 of Agbemabiese, et al.’s paper (1) or Table 1 of Hoa-Tran, et al.’s paper (Hoa-Tran TN, et al. Detection of three independently-generated DS-1-like G9P[8] reassortant rotavirus A strains during the G9P[8] dominance in Vietnam, 2016-2018. Infect Genet Evol 2020;80:104194). Also, almost invisible Figure 1 should be replaced by the revised Table 1. If the authors wish to keep Figure 1, only relevant lineages should be shown by which this reviewer means “sub-tree” so that the reader can read the strain names. However, what this reviewer recommend is to transfer the entire Figure 1 to supplementary materials.

Response to major comment 3. Table 1 was revised following your relevant suggestions. All trees were also rebuilt following strains included in table 1. We decided to maintain Figure 1 and not transfer it to supplement materials. Online publications offer the possibility to access figures at the website and zooming them in a comfortable eyes size allowing the reader to read strain names and accession numbers.

The discussion section should be reorganised in reference to the revised Table 1. Rather than what strain is most closely related to each of their G8P[4], G8P[6] and G8P[8] strains in each genome segment, similarity or distinctness of genotype/lineage constellations of four G8 strains detected in Brazil should be discussed in reference to those of G8 strains or other G types possessing the DS-1 backbone genes. From the genotype/lineage constellations of four G8 strains, it is immediately clear that while R193 has an identical genotype/lineage constellation of the DS-1-like, bovine-like G8P[8] strains possessing the DS-1 backbone genes that emerged and spread in Vietnam (such as RVN14.1149), R558 has a unique constellation in which the VP1 gene was reassorted with an yet undescribed, distinct lineage on the backbone lineage constellation identical to that of R193 and other bovine-like G8P[8] strains. This point is completely dropped from the original manuscript of the authors. Even worse is that the nature of this unique sequence is stated erroneously to be “similar to the German equine-like DS-1-like G3P[8] GER33-15 strain [63]” (lines 411-412). When this reviewer constructed a phylogenetic tree of the VP1 genes, the VP1 gene of GER33-15 was grouped clearly into lineage V just like that of R2601, and never close to that of R558. By the way the nt identity between GER33-15 and R558 was, according to this reviewer’s calculation, 92.9%, which, the authors may claim, was “the closest”, but such statement is nothing but misleading.

Response to major comment 4. Thank you so much for all the important points emphasized. Results and discussion topics were reorganized according to the revised Table 1.

In general, as to the interpretation of the phylogenetic trees, what the authors are stating is more like a repetition of what has already documented by the previous investigators (such as the artiodactyl origin of NSP2 and NSP4 genes of bovine-like G8P[8] strains), and it seems to this reviewer that the authors did not try to get any new insight (such as the one this reviewer pointed out above) that their new sequence data may potentially contain.

Response to major comment 5. All data was reanalyzed following the reviewer suggestions. Please, see results and discussion topics highlighted in yellow.

Thirdly, what surprises this reviewer is lack of reference to the work done by the Liverpool investigators who discovered the abundance of G8 strains in Africa for the first time as early as around the turn of the last century (Cunliffe NA, et al. Rotavirus G and P types in children with acute diarrhea in Blantyre, Malawi, from 1997 to 1998: predominance of novel P[6]G8 strains. J Med Virol. 1999; 57:308-312, Cunliffe NA, et al. Molecular and serologic characterization of novel serotype G8 human rotavirus strains detected in Blantyre, Malawi. Virology. 2000;274:309-320), and published about a decade ago a monumental paper similar, regarding what was done, to the authors’ current manuscript (Nakagomi T, et al. G8 rotaviruses with conserved genotype constellations detected in Malawi over 10 years (1997-2007) display frequent gene reassortment among strains co-circulating in humans. J Gen Virol 2013; 94:1273-1295). In the latter paper, it was concluded that “Malawian G8 strains are closely related to other human strains with the DS-1 genotype constellation. They have evolved over the last decade through genetic reassortment with other human rotaviruses, changing their VP4 genotypes while maintaining a conserved genotype constellation for the remaining structural and non-structural proteins.” Do the authors not challenge this earlier observation in Africa, do they? Were the previous investigators wrong or was the difference attributable to the difference in the way the G8 viruses are circulating in the two continents? This reviewer believe that the authors would make a insightful interpretation in the revised manuscript.

Response to major comment 6. Thank you for your observation. The previously studies conducted by Cunliffe et al 1999, Cunliffe et al 2000 and take into consideration. Please, see discussion topic highlighted in yellow.  

Minor comments:

The sequences that are provided to this reviewer and the ones that the authors used to draw the phylogenetic trees in Figure 1 seem different. Check if the VP7 sequence of R2437 is correct.  VP7 sequence of R2437 strain was correct for us.

Figure 1 (E) VP1:R2, The lineage that R2061, R2598, R2600 and R2597 belong to should be labelled as “Lineage V”. Corrected.

Figure 1 (I) NSP1:A2, The lineage that R193, R558, R2404, R2437, RN377, RN374, RN373, and RN361 belong to should be labelled as “Lineage IVa”. Corrected.

Round 2

Reviewer 2 Report

Title: Genomic constellation of human rotavirus G8 strains in Brazil over a 13-year period: detection of the novel bovine-like G8P[8] DS-1-like

Authors: Medeiros, et al.

This R1 manuscript was revised in accordance with this reviewer’s suggestions for improvement. While it now reads better in general, but a few awkward expressions remain that must be copy-edited before it goes to print.

Consider the title changing to …. detection of the novel bovine-like G8P[8] strains with the DS-1-like backbone.

In the same token, bovine-like G8P[8] DS-1-like strains scattered throughout the text is to be revised as bovine-like G8P[8] strains with the DS-1-like backbone.

Consider changing the order of the genes in the columns of Table 1 according to the convention; namely, VP7, VP4, VP6, VP1, VP2, VP3, NSP1, NSP2, NSP3, NSP4, and NSP5.The table will become easier to understand if such changes are made.

Author Response

Consider the title changing to …. detection of the novel bovine-like G8P[8] strains with the DS-1-like backbone.

Changed.

In the same token, bovine-like G8P[8] DS-1-like strains scattered throughout the text is to be revised as bovine-like G8P[8] strains with the DS-1-like backbone.

Changed.

Consider changing the order of the genes in the columns of Table 1 according to the convention; namely, VP7, VP4, VP6, VP1, VP2, VP3, NSP1, NSP2, NSP3, NSP4, and NSP5.The table will become easier to understand if such changes are made.

We appreciate your proposal; however, we don't believe that gene order will have an impact on data table interpretation.